# A quantitative model for the dynamics of target recognition and off-target rejection by the CRISPR-Cas Cascade complex

Marius Rutkauskas[1], Inga Songailiene [2], Patrick Irmisch[1], Felix E. Kemmerich[1], Tomas Sinkunas[2], Virginijus Siksnys [2] ✉ & Ralf Seidel [1] ✉

CRISPR-Cas effector complexes recognise nucleic acid targets by base pairing with their crRNA which enables easy re-programming of the target specificity in rapidly emerging genome engineering applications. However, undesired recognition of off-targets, that are only partially complementary to the crRNA, occurs frequently and represents a severe limitation of the technique. Off-targeting lacks comprehensive quantitative understanding and prediction. Here, we present a detailed analysis of the target recognition dynamics by the Cascade surveillance complex on a set of mismatched DNA targets using single-molecule supercoiling experiments. We demonstrate that the observed dynamics can be quantitatively modelled as a random walk over the length of the crRNA-DNA hybrid using a minimal set of parameters. The model accurately describes the recognition of targets with single and double mutations providing an important basis for quantitative off-target predictions. Importantly the model intrinsically accounts for observed bias regarding the position and the proximity between mutations and reveals that the seed length for the initiation of target recognition is controlled by DNA supercoiling rather than the Cascade structure.

CRISPR (clustered regularly interspaced short palindromic repeats)–Cas (CRISPR-associated) systems constitute adaptive RNA-guided defense systems in prokaryotes against foreign nucleic acids[1]. Cas protein effector complexes guided by the crRNA recognize and trigger subsequent cleavage of invading nucleic acids[2]. Due to their programmable cleavage specificity, effector complexes such as Cas9[3], Cas12[4] and most recently Cascade[5] were repurposed as genome editing tools in different model organisms ranging from bacteria to human cells. While the effector complexes can be addressed to target practically any unique sequence in a genome[3], they often exhibit significant promiscuity in target recognition that leads to the binding and cleavage of only partially matching DNA sequences[6,7]. Such off-targeting has been detected using high throughput techniques such as genome-wide in vivo DNA binding and cleavage studies[8–10], large on-purpose libraries for reporting DNA binding and cleavage in vivo[11,12] as well as

in vitro[13,14]. Off-targeting can result in highly undesired and unpredictable genetic rearrangements which is particularly problematic for therapeutic applications[6].

The recent development of engineered effector variants[15–17] can reduce but not abolish off-targeting[16,18]. A frequently used complementary approach to prevent off-targets are in silico off-target predictions that promise to identify crRNAs with the least promiscuity[19–21]. Prediction tools use typically heuristic scoring functions that try to reproduce sequence and mismatch position patterns from high throughput studies. Though many strong off-target sites are correctly predicted, a considerable fraction of weaker off-targets remains undiscovered by the algorithms[9,22], such that off-targeting persists to be a challenging problem of CRISPR-Cas technologies. Furthermore, these algorithms do not provide quantitative measurable parameters and cannot predict how off-targeting changes with

[1]Peter Debye Institute for Soft Matter Physics, Universität Leipzig, 04103 Leipzig, Germany. [2]Institute of Biotechnology, Life Sciences Center, Vilnius University, Saulėtekis ave. 7, Vilnius 10257, Lithuania. ✉e-mail: siksnys@ibt.lt; ralf.seidel@physik.uni-leipzig.de

altered conditions, such as the local genomic supercoiling or the enzyme concentration.

Along with extensive characterization of off-targeting, considerable mechanistic insight into the target recognition process by CRISPR-Cas effector complexes has been obtained from biochemical, structural and single-molecule studies. A converging theme emerged (Fig. 1a) in which an effector complex first scans duplex DNA for a short complex-specific protospacer adjacent motif (PAM). Upon PAM recognition, it initiates base pairing between the crRNA and the PAM adjacent bases of the DNA target strand. The RNA-DNA heteroduplex can then reversibly expand expelling the non-target DNA strand and forming a triple-stranded R-loop structure. Upon full-length R-loop formation up to the PAM-distal end, a conformational change occurs that licenses DNA degradation[23–25]. For Cascade the latter comprises a global sliding of the Cse1–Cse2 filament[26,27] that locks the R-loop in a highly stable conformation[25,28] and allows the recruitment of the Cas3 nuclease (step iv in Fig. 1a). The actual target recognition is, however, a strand displacement reaction between the involved nucleic acid strands (steps ii and iii in Fig. 1a), in which the effector complex acts only as a sensor for the R-loop progression. Mismatches can be considered as energy barriers during the reversible R-loop expansion promoting its collapse[28]. The mismatch strength has hereby shown to be biased with respect to the position; PAM-proximal mismatches

within the so-called seed region have been shown to impose stronger inhibition of R-loop formation compared to distal mismatches[8,29]. In general, the reversible nature of R-loop expansion and collapse in competition with irreversible locking or cleavage impose a kinetic rather than a typical "sticky", i.e., affinity-based, target recognition mechanism[28,30,31].

Despite an increasing detailed mechanistic understanding of target recognition by CRISPR–Cas complexes, the wealth of mechanistic knowledge has until recently[32] not been exploited for off-target predictions nor, despite widely suggested, been applied to quantitatively understand the targeting dynamics in mechanistic studies[30,31,33].

To establish such a link, we use here single-molecule DNA twist measurements to comprehensively quantify the dynamics of R-loop formation by the Cascade complex from *Streptococcus thermophilus*. Importantly, we resolve the transient R-loop sub-states on single- and double-mismatched DNA targets which remained hidden for other methods including high-throughput off-targeting measurements[13,14]. We show that the observed dynamics can be quantitatively modeled by describing R-loop formation as a random walk model in a simplified one-dimensional free energy landscape. The model was adapted from previous descriptions of protein-free strand displacement reactions in dynamic DNA nanotechnology[34,35], which have recently been introduced to the CRISPR–Cas field[31,32]. Importantly, our modeling (i)

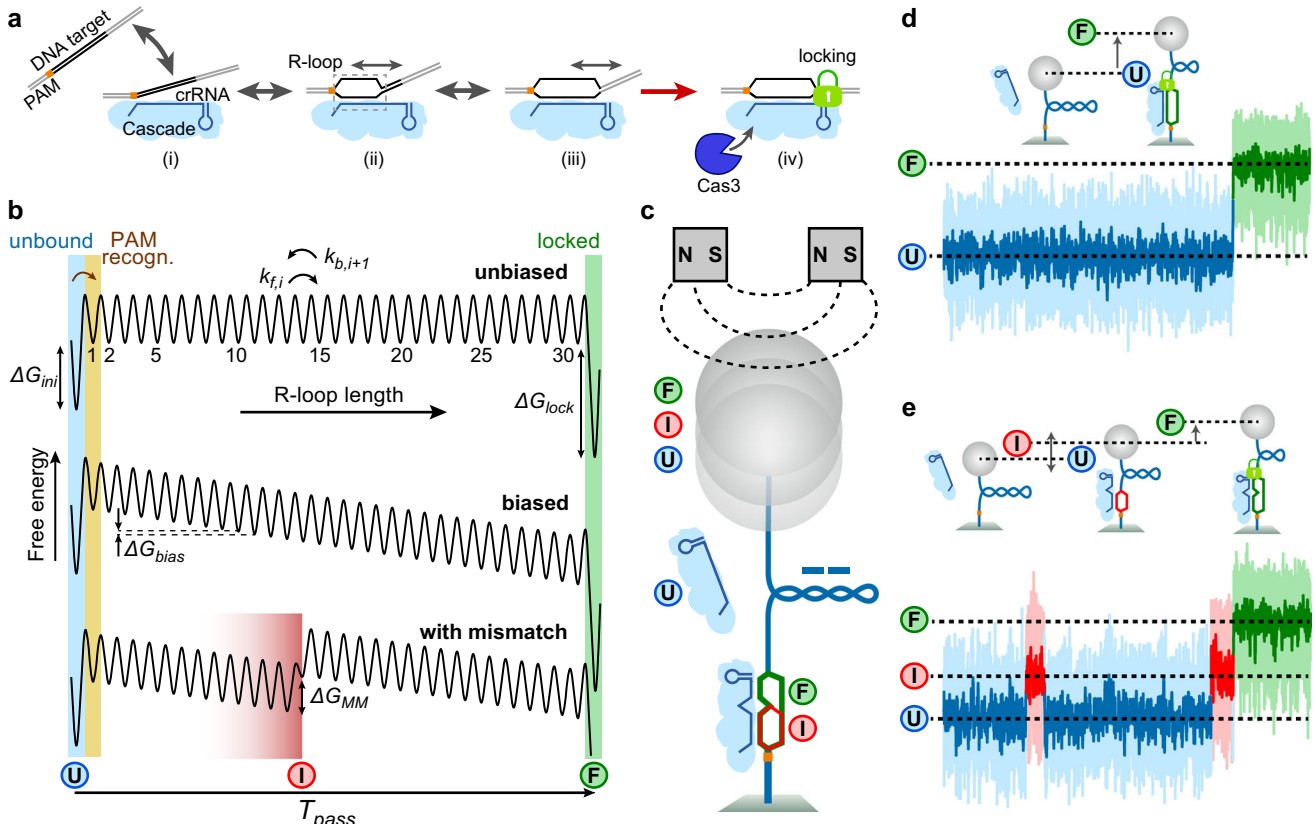

**Fig. 1 | Random-walk model for target recognition by Cascade. a** Target recognition starts with PAM binding (i) followed by R-loop priming adjacent to the PAM. The R-loop can expand in a reversible random-walk like fashion towards the PAM distal end of the target (ii). After reaching the PAM distal end (iii), a conformational change is induced that locks the R-loop in a stable conformation and enables Cas3 recruitment and cleavage of the DNA (iv). **b** Simplified one-dimensional free energy landscapes for R-loop formation. Local energy states are the unbound state (*U*), the R-loop states of different length and the locked full R-loop (*F*). R-loop expansion and shrinkage occurs in successive forward or backward steps to either adjacent state. Shown free energy landscapes correspond to R-loop formation without an energy bias (top), with a negative bias as expected for

negative supercoiling or due to favorable molecular interactions (middle) and with a mismatch at position 14, which introduces a local energy barrier (bottom). In the latter case an additional dynamic intermediate R-loop state (*I*) is introduced. **c** Real-time detection of the R-loop dynamics on negatively supercoiled DNA using magnetic tweezers. R-loop expansion progressively absorbs part of the introduced supercoiling, which results in a DNA length increase. **d** R-loop formation on a fully matching target is seen as a single abrupt DNA length increase corresponding to the transition from the unbound to the full-R-loop state. **e** R-loop formation on a target with a single internal C:C mismatch at position 17 occurs via additional transitions to and from the unstable intermediate state until the full locked R-loop is formed.

provides direct evidence that R-loop expansion down to local sub-states follows a random-walk process (ii), shows that the single-base pair stepping of R-loop expansion occurs at a sub-millisecond time scale, (iii) returns absolute free energy penalties imposed by different mismatches, (iv) quantitatively predicts the non-trivial dependence of R-loop formation on the proximity between multiple mismatches and (v) reveals that the length of the seed region in Cascade is a function of the applied supercoiling rather than a structural property.

Overall, our findings establish an important mechanism-based approach for the prediction of off-targeting by Cascade and potentially other genome engineering tools. We furthermore provide quantitative insight into how off-targeting depends on DNA super-coiling, which in eukaryotes has been found to be highly locus- and gene specific[36,37].

## Results

### Modeling the R-loop dynamics of Cascade
Previous investigations of the target recognition by Cascade[14,28,31,38] and other effectors[13,24,32] strongly suggested that following PAM binding, a dynamic R-loop structure gets nucleated which stochastically grows and shrinks with single base-pair steps[31,39]. When the R-loop expands until the PAM-distal end, a locking transition and/or cleavage is triggered (Fig. 1a). Recently, models were developed that describe protein-free DNA strand displacement reactions and R-loop formation as a random walk in a one-dimensional free energy landscape[32,34,35,40,41]. These models differ mainly in their system-specific energy landscapes that allow local rate variations. To develop and test a quantitative model for Cascade, we constructed a suitable one-dimensional free energy landscape for R-loop formation with the R-loop length as reaction coordinate (Fig. 1b). Cascade starts initially in the unbound state ($U$) corresponding to an R-loop of zero length. Upon PAM binding an R-loop of 1 bp length shall be nucleated for which a free energy penalty $\Delta G_{\mathrm{ini}}$ is considered. In a first iteration we assumed that the free energy for increasing R-loop lengths is constant since for each additional base-pair of the heteroduplex a base pair of the DNA-duplex needs to be disrupted (Fig. 1b, top). After full R-loop formation of 32 bp for Cascade, the R-loop enters the locked state (state $F$) associated with a decrease in free energy by $\Delta G_{\mathrm{lock}}$. Negative supercoiling has been shown to assist R-loop formation[25,28]. This provides a constant negative bias $\Delta G_{\mathrm{bias}}$ for increasing R-loop lengths (Fig. 1b, middle). Also, protein contacts to the R-loop may contribute an additional favorable or unfavorable bias that we assume to be constant throughout the R-loop. For each mismatch that acts as a barrier for R-loop expansion, a local mismatch penalty $\Delta G_{\mathrm{MM}}$ was introduced that shifts all steps behind the mismatch equally upwards (Fig. 1b, bottom). A mismatch introduces thus a dynamic intermediate R-loop state $I$, which extends over a few base pairs before the mismatch due to the random-walk nature of the R-loop (Fig. 1b, bottom). In absence of a free energy bias, the kinetic barriers between all transitions were assumed to be identical and described by the unbiased single base-pair stepping rate $k_{\mathrm{step}}$. In case of bias, as given by the free energy landscape, rate alterations were described using Arrhenius' law (see Methods for details). With these few assumptions a fully determined linear rate model was obtained. Mean transition times between R-loop states could be calculated by solving the first passage problem for this model (see Supplementary Note 1). This allowed us to obtain the mean R-loop formation time but also the rates for transitions between different R-loop intermediate using $\Delta G_{\mathrm{ini}}, \Delta G_{\mathrm{bias}}, k_{\mathrm{step}}$ and in case of mismatches one or several different values for $\Delta G_{\mathrm{MM}}$ as the only free parameters.

### Random walk describes dynamics of intermediate R-loops
To test the applicability of the random walk model, we set out to comprehensively quantify the R-loop dynamics of Cascade using single-molecule DNA twisting experiments[25,28,38]. Surface-grafted DNA molecules tethering a magnetic bead on their free end were stretched

vertically in a magnetic tweezers apparatus[42,43]. Negative supercoiling introduced by rotating the tweezers magnets provided a DNA length reduction due to the formation of writhe[28,44,45] (Fig. 1c). DNA unwinding due to R-loop formation by Cascade absorbs part of the introduced supercoiling and causes a DNA length increase that is proportional to the unwound base pairs[43] (Fig. 1c, d). This allows to resolve full, locked R-loop formation as well as dynamic R-loop intermediates[24,28,33] (Fig. 1d, e).

We first quantified the dwells of intermediate R-loops of different lengths. We used DNA targets containing a limited number of match-ing base pairs adjacent to the PAM (from 8 to 22 bp) with the remaining base pairs being mismatched. This way only transitions between the $U$ and the $I$ states were observed (Fig. 2a and Supplementary Fig. 3a). We furthermore applied different negative supercoiling levels, given as mechanical torque $\tau$, that were controlled by the applied stretching forces (see Methods). Natural superhelical densities $\sigma$ in E. coli cells are in the range of $-0.06$ and $-0.029$ corresponding to torques between $-8.9$ and $-4.3\,\mathrm{pN\,nm}$[46,47] (Methods). Qualitative inspection of the obtained trajectories revealed that the dwell in the $I$ state increased with increasing negative supercoiling (Fig. 2b) and increasing length of the R-loop intermediate (Fig. 2c). This intuitively agrees with the free energy landscape of the random-walk model, since increased bias and length lower the free energy of the $I$ state and thus increase the energy barrier for a diffusive return to the $U$ state (Fig. 2f) leading to increased occupancies when modeling the $I$ state (Fig. 2f). Quantitative analysis of the dwells in both states provided the R-loop formation rates $k_1$ (Fig. 2d) and collapse rates $k_2$ of the $I$ state (Fig. 2e). $k_1$ was rather independent of the R-loop length and the applied supercoiling. In contrast, the R-loop collapse rate $k_2$ was strongly torque- and R-loop length-dependent and varied over three orders of magnitude. A global fit of the random-walk model to all collapse rates correctly described both the large spread of the rates between the different R-loop lengths as well as the torque dependence for a given R-loop length. This pro-vides direct support that R-loop expansion and retraction follow a random-walk mechanism. Remarkably, the fitting used the unbiased single base-pair stepping rate $k_{\mathrm{step}} = 1900 \pm 100\,s^{-1}$ as the only free parameter. Single base-pair steps during R-loop formation thus occur on the sub-millisecond time scale. Beyond the rates, also the occu-pancies of the $U$ and $I$ states were correctly described including their torque and length dependencies (see histograms in Fig. 2b, c).

### R-loop dynamics at single mismatches provides mismatch penalties
We next studied the R-loop dynamics on targets containing a single mutation at about half the target length and six consecutive mis-matches at the PAM-distal end to prevent R-loop locking[25,28]. On such a target, the R-loop fluctuated between three states – the $U$, the $I$ and a dynamic $F^*$ state of a maximum length of 26 bp (Fig. 3a). Testing the three possible mismatches C:C, C:T, and C:A at position 17 (counting from the PAM) revealed that the mismatch type strongly influenced the transition rates and the occupancies of the three states (Fig. 3b). Furthermore, these parameters were influenced by increasing negative torque, where the $F^*$ state became increasingly populated at the expense of the $U$ state (Supplementary Fig. 4a). We quantitatively analyzed the four rates that describe the transitions between adjacent states (Methods, Supplementary Fig. 5). This revealed that the rates and their torque dependences were largely independent of the mis-match type except $k_3$ that described the mismatch passage from the $I$ to the $F^*$ state (Figs. 3d, e). Generally, rates describing R-loop expan-sion ($k_1, k_3$) increased with increasing negative torque, while rates describing R-loop retraction ($k_2, k_4$) were found to decrease. For a given mismatch, we applied a global fit to the torque dependence of all four rates (solid lines in Figs. 3d and 3e) yielding good agreement with the data. Consistently, expected occupancies of the $U$, $I$, and $F^*$ cal-culated from the best fit parameters were also in agreement with the

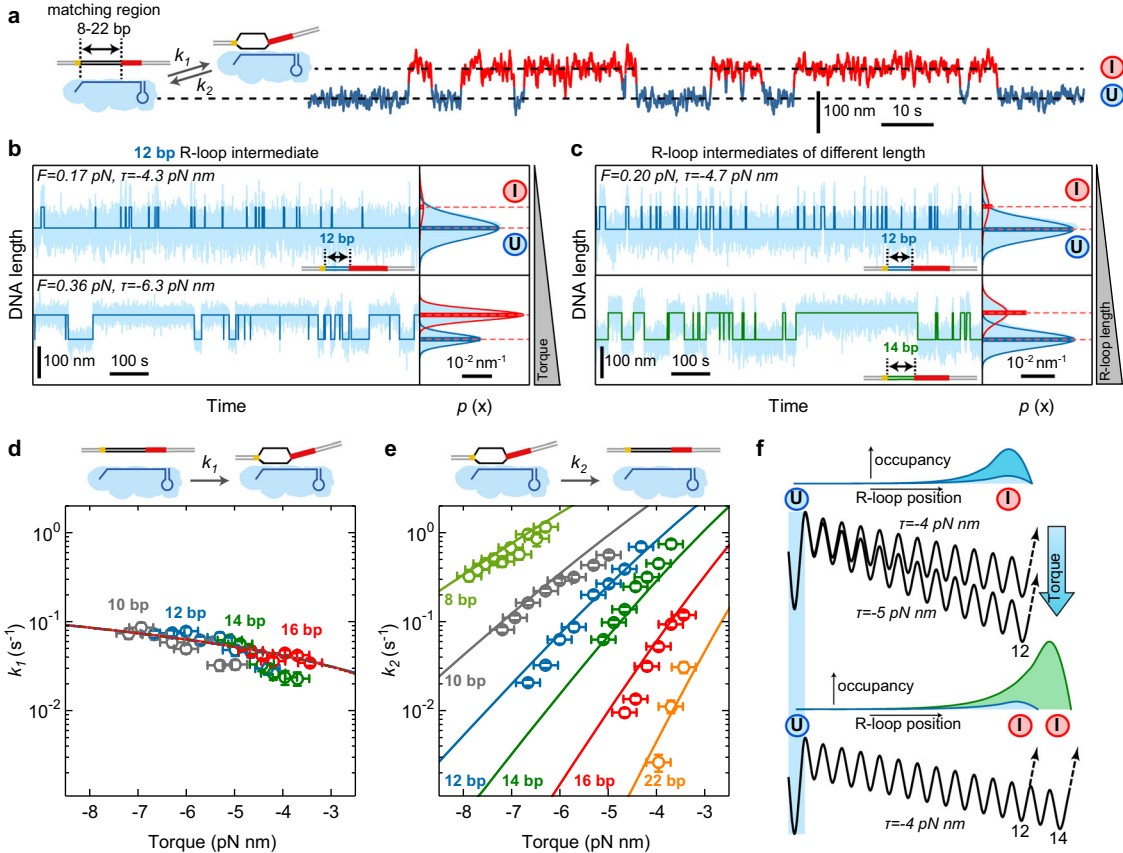

**Fig. 2 | Formation and collapse rates of R-loop intermediates of different lengths. a** Sketch of the DNA construct containing a limited number of matching PAM-proximal bases such that only R-loop intermediates form (left). Example trajectory of the transitions between unbound state (*U*) and intermediate R-loop (*I*) for a target with 12 matching bases. **b** Formation and collapse of R-loop intermediates recorded at different torques τ for a target with 12 matching bases. Shown are the recorded DNA length smoothed to 7.5 Hz (light blue) and a two-state approximation of the trajectory (dark blue). Solid lines in the histograms on the right represent Gaussian fits to the 2 different states, while horizontal dashed lines indicate the average DNA length of each state. Bars represent theoretically

predicted occupancies using best fit parameters from (**d**) and (**e**). **c** Formation and collapse of R-loop intermediates of different lengths at a torque of −4.7 pN nm. **d**, **e** Measured mean formation (**d**) and collapse (**e**) rates of the R-loop intermediates as a function of torque for different lengths of the matching region (open circles). A global fit to the data is shown as solid lines. Rate errors are given as SEM; torque errors correspond to 0.25 pN nm. Precise sample sizes are given in the Supplementary Table 6. Source data are provided as a Source Data file. **f** Occupancies of the *I* state for two different negative torques (upper panel) and R-loop lengths (lower panel) together with the corresponding free energy landscapes.

measurements (Fig. 3b and Supplementary Fig. 4a right panels). The obtained values for $k_{step}$ (Supplementary Fig. 4b) and $\Delta G_{ini}$ (Supplementary Fig. 4c) were mismatch-independent while the mismatch penalty $\Delta G_{MM}$ (Fig. 3c left panel, Supplementary Table 1) was strongly mismatch-dependent. This is consistent with the intuitive expectation that the mismatch type influences only the corresponding penalty but not the other parameters. Applying different Cascade concentrations revealed a linear dependence of the initial R-loop intermediate formation rate $k_1$ on the concentration while leaving the other rates unchanged (Supplementary Fig. 4d–g) in agreement with concentration-independent values for the standard free energy of R-loop initiation and the other model parameters (Supplementary Fig. 4f). Altogether, the three-state dynamics over a single mismatch was well described by the random walk model.

Next, we investigated how $\Delta G_{MM}$ depended on the mismatch position. When keeping the same mismatch type including the same nearest-neighbor base pairs, $\Delta G_{MM}$ should be largely position independent if nucleic acid thermodynamics would dominate. To test this, we produced different Cascade complexes with a CCC stretch at different positions in the crRNA (Supplementary Tables 3 and 4) allowing a corresponding introduction of a C:C mismatch with two adjacent G:C base pairs (see sketch in Fig. 3c, right panel). Recording and fitting the torque dependence of the different transition rates for these

complexes revealed that $\Delta G_{MM}$ was within error invariant for mismatch positions between 11 and 17 bp that could be experimentally accessed (Fig. 3c right panel, Supplementary Table 1). We note that mismatch barriers were not observable at positions 6, 12, 18, 24, and 30[8,28] due to the disrupted base pairing in the crRNA–DNA hybrid at these positions[48].

## Seed length is dependent on supercoiling

A mismatch penalty that is rather independent of the mismatch position seems to contradict the larger impact of PAM-proximal mismatches in the seed region compared to PAM-distal mismatches observed in vivo[8,29] and in vitro[14,28]. To resolve this apparent contradiction, we expanded the range of mismatch positions from 5 to 21 bp. Since for many of these targets intermediates were too short-lived to be observable, we measured only the time for full R-loop formation (state *F*) using a full-length target that supported locking (Fig. 4a and Supplementary Fig. 3b–e). In agreement with previous reports[28,29], R-loop formation was slower for targets with PAM-proximal mismatches compared to PAM distal mismatches and the WT target (Supplementary Fig. 6a, b). We determined the mean R-loop formation time as a function of torque for the WT target and the single mismatch targets (Fig. 4b). R-loop formation for the WT target was little dependent on torque in the applied range. The R-loop formation times for

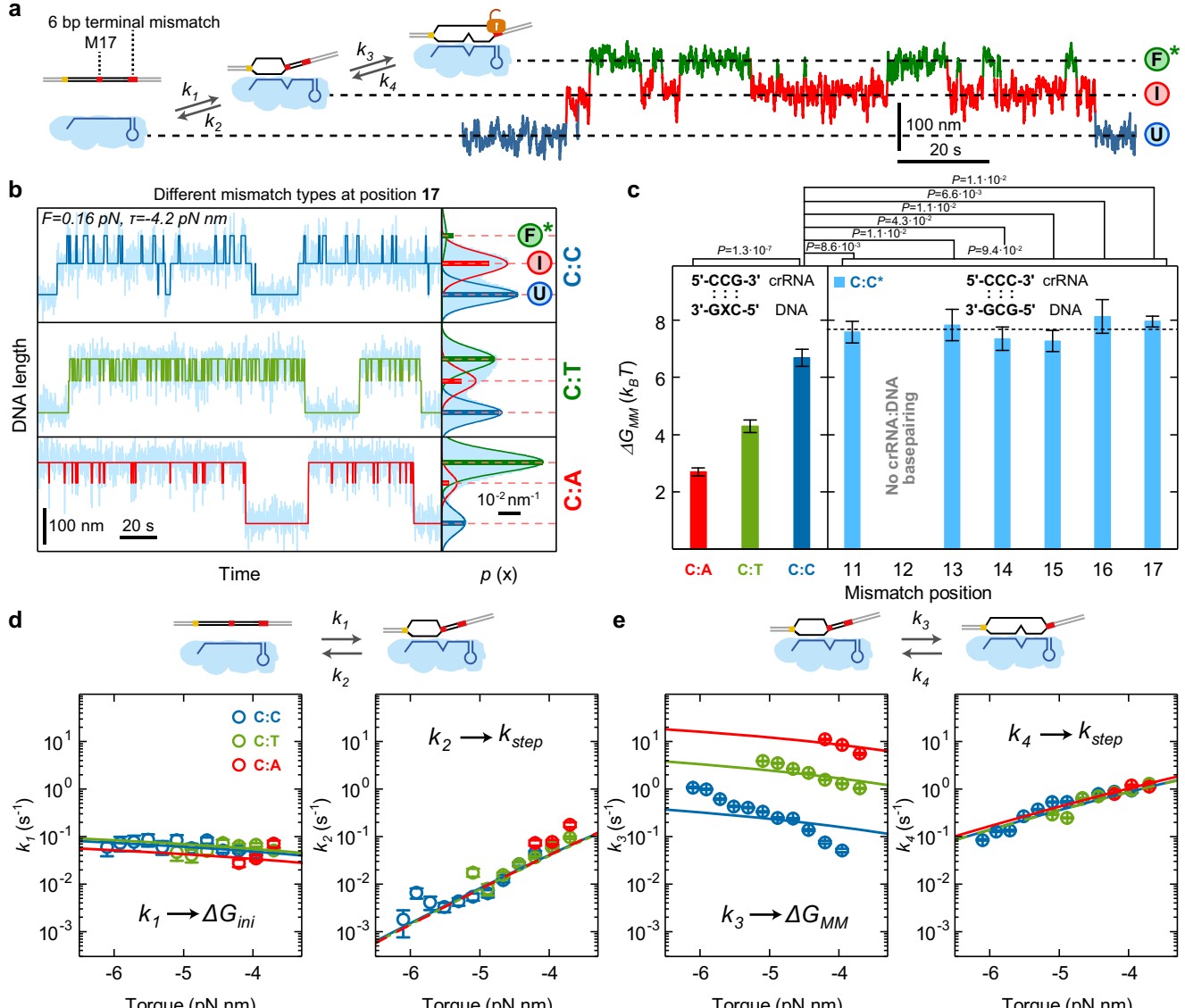

**Fig. 3 | R-loop dynamics on locking-deficient targets containing a single internal mismatch. a** Sketch of the DNA target with mismatched bases shown in red and the possible R-loop states—the unbound state ($U$), the intermediate R-loop ($I$) and the almost full but unlocked R-loop ($F^*$). Sequential transitions between these states can be described by 4 different rate constants ($k_1$ to $k_4$) as indicated. Example trajectory of the transitions between the different states measured for a C:C mismatch. **b** Trajectories and histograms of the DNA length recorded for targets with either a C:C, C:A, or a C:T mismatch at a torque of 4.2 pN nm (light blue). 3-state approximations of the trajectories are shown in dark blue (C:C mismatch), green (C:T mismatch) or red (C:T mismatch). Solid lines in the histograms represent Gaussian fits to the 3 different states, while horizontal dashed lines indicate the average DNA length of each state. Bars represent theoretically predicted

occupancies using the parameters shown in Supplementary Table 1. **c** Mismatch penalties for the different mismatches at position 17 (left panel) and of the C:C mismatch at different positions (right panel) obtained from fitting the transition rates presented in (**d**) and (**e**). Error bars correspond to SD of the fit parameter (67% confidence interval). **d** Experimental transition rates between the $U$ and the $I$ states as a function of torque for C:C, C:T, and C:A mismatches are shown as blue, green, and red circles, respectively. **e** Experimental transition rates between the $I$ and the $F^*$ state as a function of torque. The solid lines shown in (**d**) and (**e**) correspond to a global fit to all four transition rates for a given mismatch. Error bars in all plots correspond to SEM. Precise sample sizes are given in the Supplementary Table 6. For statistical testing on-way ANOVA was used. Source data are provided as a Source Data file.

single-mismatch targets decreased, however, strongly in a non-linear fashion with increasing negative torque and finally plateaued at the WT level (Fig. 4b and Supplementary Fig. 6d). The torque required for reaching the WT level was changing monotonously with mismatch position. We applied a global fit of the random walk model to the data using $\Delta G_{ini} = 8.5\,k_B T$ and obtained agreement with the experimental results (Fig. 4b and Supplementary Fig. 6d). The consideration of single mismatch targets alongside the WT target in this data set as well as the highly non-linear torque dependence allowed furthermore to probe a potential intrinsic bias $\Delta G_{bias}$ of the free energy landscape in absence of torque. Global fitting with a free non-zero bias provided as

best fit parameters $\Delta G_{MM} = 6.9 \pm 0.5\,k_B T$, $k_{step} = 2000 \pm 900\,\mathrm{s}^{-1}$ as well as $\Delta G_{bias} = 0.14 \pm 0.01\,k_B T/bp$. The positive value of $\Delta G_{bias}$ reveals that in absence of torque the free energy landscape has a small upward bias (Supplementary Fig. 6g) corresponding to an apparent torque of -1 pN nm which is much smaller than base pairing energies and mismatch penalties. Inclusion of the determined $\Delta G_{bias}$ into fits of the previous data were mainly compensated by changes of $k_{step}$ while the obtained free energy values became only slightly reduced (see Supplementary Table 2).

To explore the impact of seed mismatches in more detail, we plotted the measured R-loop formation times for selected torque

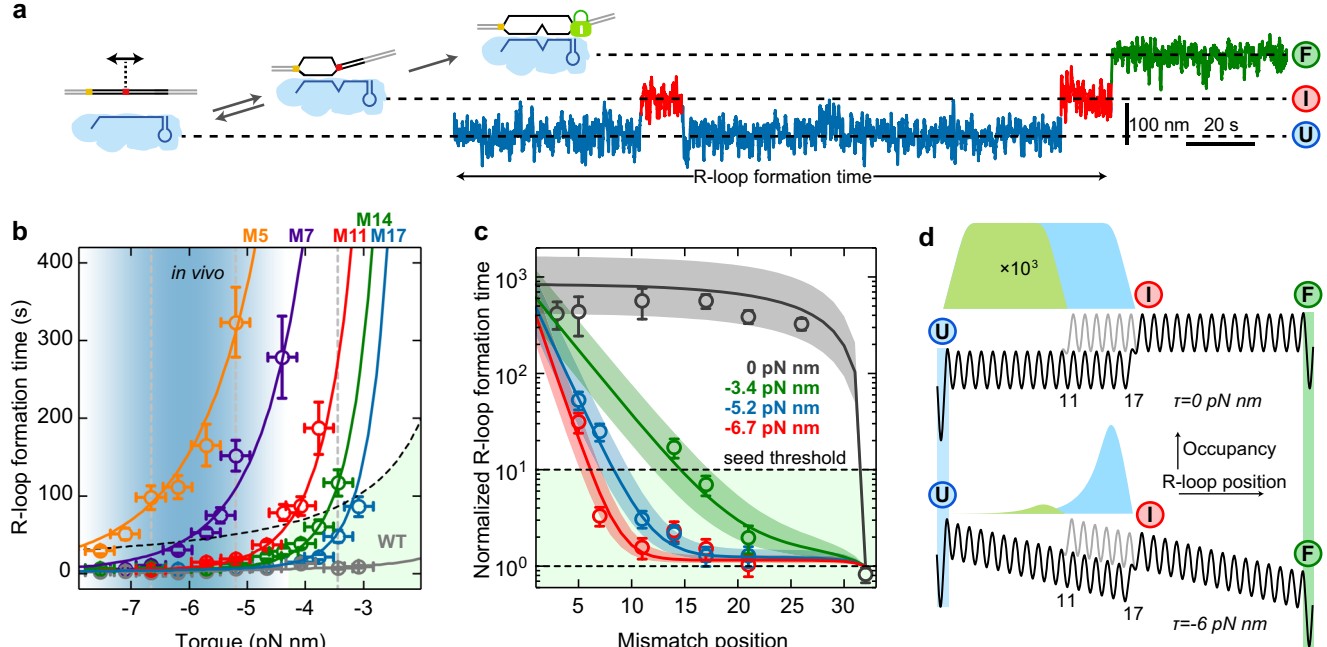

**Fig. 4 | Locked R-loop formation on targets with single internal mismatches.**
**a** Sketch of the DNA construct containing a single C:C mismatch at various positions along the target (left). Example trajectory of full R-loop formation for a mismatch at position 17 after applying negative supercoiling. The full R-loop is formed after multiple intermediate state events. **b** Torque dependence of the R-loop formation time for different mismatch positions. Circles represent mean times from >27 events collected on at least two molecules. Solid lines represent a global fit of the random-walk model to the data. The light blue gradient marks the torque range estimated in vivo (see Methods). The light green area corresponds to a less than 10-fold increase of the R-loop formation time compared to the WT. Errors of the

formation time are SEM and of the torque 0.25 pN nm. **c** Normalized R-loop formation time with respect to the WT (open circles, obtained from data in (**b**), as well as Supplementary Fig. 6a, f) as a function of the mismatch position for three different torques (corresponding to the vertical dashed lines in (**b**)). Solid lines show the predictions of the random-walk model using the fit parameters from (**b**). Colored areas represent single confidence intervals of the fits. Error bars correspond to SEM. **d** Comparison of the occupancies of the intermediate R-loop state $I$ for mismatches at positions 11 and 17 in the absence (upper panel) and presence (lower panel) of negative torque. Precise sample sizes are given in the Supplementary Table 6. Source data are provided as a Source Data file.

values as a function of the mismatch position (Fig. 4c). Using a 10-fold increased R-loop formation time compared to the WT target as a hypothetical threshold for a seed mismatch, we observed that the length of the seed region decreased with increasing torque from ~6 bp at −6.7 pN nm to ~15 bp at −3.4 pN nm. Predicting the R-loop formation time in absence of an external torque suggested that in this case, the seed region would cover almost the entire target sequence (see black line in Fig. 4c). To verify the prediction, we set up a fluorescence bulk solution assay based on a donor-quencher pair at the PAM distal end (Supplementary Fig. 6e) and measured the R-loop formation kinetics in the absence of supercoiling for different mismatch positions (Supplementary Fig. 6f). The extracted mean times of R-loop formation confirmed the theoretical prediction (Fig. 4c) demonstrating that the seed region in Cascade is mainly a product of the applied supercoiling, i.e., the bias of the free energy landscape. This can be intuitively understood by considering the different occupancies of the intermediate R-loop state $I$ for PAM-proximal and distal mismatches, which determines the full R-loop formation rate. In absence of a bias, all R-loop lengths up to the mismatch are energetically equal and thus equally populated, which supports similar R-loop formation times (Fig. 4d upper panel). In presence of a negative bias, the $I$ state of a PAM-proximal mismatch is energetically higher with respect to the $U$ state and thus less populated compared to a PAM-distal mismatch (Fig. 4d lower panel). This provides a comparably slower transition to the $F$ state for the PAM-proximal mismatch.

## Intermediate R-loop dynamics in the presence of two mismatches
After verifying the random walk model for targets with single mismatches, we next tested whether it can be directly applied to describe

the R-loop dynamics in presence of two mismatches. We produced double point mutants with the first mismatch located at positions 11, 13 or 14 and the second at position 17 within the locking deficient target (PAM-distal mismatches at positions 27-32). In this case R-loops could fluctuate between four possible states: $U$, the intermediate states before each mismatch $I$ and $I^*$ as well as $F^*$ (Fig. 5a). For the 11–17 double mismatch target, the $I$ and the $I^*$ states could be distinguished, but the transitions between the states were too fast to be resolved for the closer mismatch spacings (Fig. 5b). State occupancies and transition dynamics were again torque-dependent (Supplementary Fig. 7c). Using the best-fit parameters from the single mismatch experiments, the random walk model predicted the measured state occupancies remarkably well (Fig. 5b and Supplementary Fig. 7c right panels). A 4-state approximation of the recorded trajectories for the 11–17 double mismatch substrate (Supplementary Fig. 7c, bottom panel) allowed to extract the six transition rates between subsequent states. Consistently, we obtained agreement for the extracted rates from the measurements and from Brownian dynamics simulations based on the model predictions (Supplementary Fig. 7a, b, d). Of note, the $F^*$ state was less frequently visited as closer the two mismatches were positioned (Fig. 5b) revealing that the mismatch proximity influences the formation of the full R-loop.

## Proximity between double mismatches strongly influences R-loop formation
To investigate the influence of the proximity between two mismatches in detail, we studied locked R-loop formation on double C:C mismatch targets without terminal mismatches (Fig. 6a). Predictions by the random walk model showed that a strong inhibition is obtained when combining two PAM-proximal mismatches (brown areas in Fig. 6b,

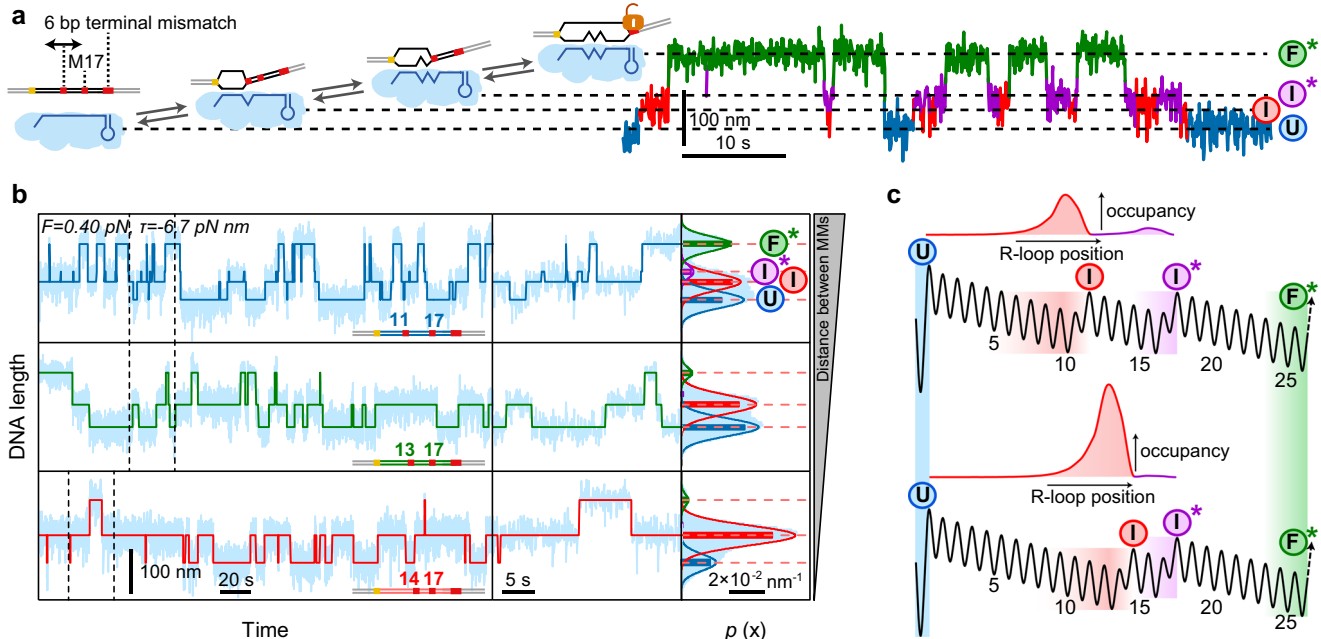

**Fig. 5 | R-loop dynamics on locking-deficient targets containing two internal mismatches. a** Sketch of the DNA construct with mismatched bases shown in red and the possible R-loop states: the unbound state (*U*), two intermediate R-loops (*I* before the first and *I\** before the second mismatch) and the almost full but unlocked R-loop (*F\**). Sequential transitions between these states can be described by six different rates. Example trajectory of the transitions between the different states measured for a C:C mismatches. **b** Trajectories and histograms of the DNA length recorded for targets containing C:C mismatches with varying proximities at a torque of −6.7 pN nm (light blue). Four-state and three-state (in case *I* and *I\** states

were indistinguishable) approximations of the trajectories are shown as dark lines. The central panel shows enlarged views of the regions between the dashed lines. Solid lines in the histograms represent Gaussian fits experimental histograms, while horizontal dashed lines indicate the average DNA length of each state. Bars represent theoretically predicted occupancies using the parameters shown in Supplementary Table 1. **c** Model of free energy landscapes for R-loop formation for the 11–17 (top) and the 14–17 (bottom) double mismatch targets together with the normalized occupancies of the *I* and *I\** states obtained from the model. Source data are provided as a Source Data file.

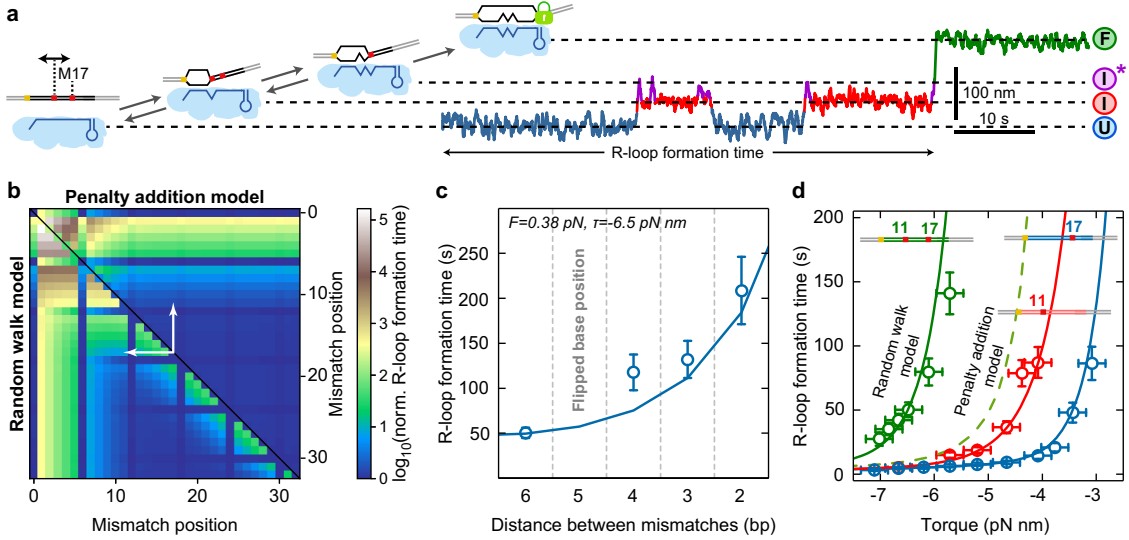

**Fig. 6 | Locked R-loop formation on targets containing double mismatches. a** Sketch of the DNA construct containing a first C:C mismatch at positions varying from 11 to 15 and a second C:C mismatch at position 17 (left). Example trajectory of full R-loop formation with the first mismatch located at position 11 containing two intermediate states *I* and *I\**. **b** Predicted R-loop formation time for double mismatches relative to the matching target using the penalty for a C:C mismatch and torque of −6.5 pN nm. Either the penalty addition model (upper right half) or the random walk model (lower left half) was applied. For mutations at positions 6, 12, 18, 24, and 30 no penalties were considered due to the disrupted base-pairing in the crRNA–DNA hybrid. White arrows indicate the range of considered mismatch distances between positions 11 and 17. **c** Experimental R-loop formation times on

targets with double mismatches with a variable position of the first mismatch and a fixed position of 17 for the second mismatch. The solid line represents the model prediction using the parameters from the single mismatch experiments (see Supplementary Table 1). Error bars correspond to SEM. **d** R-loop formation time as a function of torque for targets with single mismatches at positions 11 and 17 and the target containing both mismatches. Solid lines represent the model prediction using the parameters from the single mismatch experiments (see Supplementary Table 1). The dashed line represents the prediction of the penalty addition model for the target with both mismatches. Error bars correspond to SEM; torque errors correspond to 0.25 pN nm. Precise sample sizes are given in the Supplementary Table 6. Source data are provided as a Source Data file.

lower left corner) while a weak inhibition is obtained when combining two PAM-distal mismatches (blue areas in Fig. 6b). However, a considerable inhibition is also obtained when combining two PAM-distal mismatches that are in close proximity (green tail along the diagonal in Fig. 6b). This influence of the proximity between mismatches was only obtained for the random walk model but not for a simple addition of apparent free energy penalties (Fig. 6b, upper right, Supplementary Fig. 8c, d) as applied in heuristic scoring schemes[11].

We next determined R-loop formation times as a function of torque for a second mismatch at the fixed position 17 and distances to the first mismatch ranging from 2 to 6 bp (Supplementary Fig. 8a, b). The mean R-loop formation time at a given torque increased strongly with decreasing distance between the two mismatches (Fig. 6c). This dependence on the mismatch proximity was quantitatively described by the random-walk model using the parameters from single-mismatch experiments (Fig. 6c, d). Intuitively, the proximity dependence can be understood by considering that for close mismatches the $I$ state is energetically lower than for distant mismatches, such that the occupancy of the $I^*$ is lowered at the expense of the $I$ state (Fig. 5c). This in turn provides less frequent transitions to the $F$ state for proximal compared to distal mismatches.

## Discussion

In this study, we presented a highly comprehensive investigation of the R-loop dynamics within a CRISPR-Cas type I Cascade effector complex. Using direct measurements of the DNA untwisting during R-loop formation, we could uniquely resolve multiple R-loop intermediates on mutated DNA targets and carefully quantify their behavior as a function of the positions and types of the formed mismatches.

Importantly, the diverse data set could be consistently described by modeling the R-loop formation process as a random walk in a simplified one-dimensional free energy landscape. Previous work modeled the impact of the mismatch position on the global strand-displacement or R-loop formation reaction[32,35] and reproduced general observed dependencies. Here, we demonstrate in great detail that the entire R-loop formation dynamics, including the substates, follows a random walk mechanism. Thus, the forming R-loop samples the continuum of available R-loop lengths, and unlocked R-loop intermediates dynamically extend over a range of different lengths (Fig. 2f). The time scale of the single base-pair R-loops steps was found to be in the sub-millisecond range (Supplementary Tables 1 and 2). This allows for a reversible, highly dynamic sampling of the sequence space during target search in which mismatches stall the R-loop formation process and promote its collapse (Fig. 4a). Combined with the irreversible locking step, this gives rise to a kinetic target recognition process in contrast to typical binding-affinity-based mechanisms[30]. In principle, given sufficient time, any target can become recognized. This sets important constraints when discussing specificities of effector complex variants[15,16].

Overall, our data is well described by a rather simple form of the free energy landscape which in absence of mismatches exhibits just a constant supercoiling-dominated bias (Fig. 1b). This strongly suggests that a global and rather uniform bias will be the dominating feature of the free energy landscape of R-loop formation by Cascade. Some deviations between data and model do occur, however (see e.g., Fig. 2e). The relative rate deviations appear small (typically smaller than a factor of two) compared to the orders of magnitude covered and are not too surprising given the simplicity of the applied landscape. In addition to experimental errors, we mainly attribute the deviations to sequence- and/or structure-dependent local alterations of the free energy landscape from the simple model. As noted before, the crRNA-DNA hybrid displays disrupted base pairing every 6 bp[48]. A reduced stabilization of the unpaired DNA base and the absence of base stacking interactions may make these positions energetically less favorable, such that the global downhill bias of the free energy

landscape will likely exhibit a periodic modulation. This may further be enhanced by the 6-bp periodicity of the Cas7 backbone, e.g., due to periodically recurring interaction sites with the DNA duplex during R-loop expansion[26]. This modulation does not appear to have a major impact on the observed trends of the data (see e.g., the R-loop length dependence in Fig. 2e) but may cause smaller alterations. Nonetheless, the modulation of the free energy landscape may give rise to short-lived kinetic R-loop intermediates every 6-bp which would be in agreement with a kinetically metastable partial R-loop structure of Cascade found at low temperature[26].

An important finding of our data and modeling is that the seed sequence observed for all R-loop forming CRISPR-Cas effector complexes[49] is at least partially a biophysical consequence of the external DNA supercoiling. In absence of supercoiling, the free energy landscape of Cascade was found to be a little biased. This provided a rather position-independent impact of the mismatches which can be interpreted as an 'extended seed' for Cascade targeting (Fig. 4c). In contrast, at sufficient negative bias of the free energy landscape induced by supercoiling, the target recognition was only little affected by PAM-distal mismatches which strongly limited the seed extension. The observed torque-dependence of the seed length (Fig. 4b) can explain observations of a short well-defined seed region in the cellular environment[8,29] in contrast to more relaxed seed conditions in vitro in absence of supercoiling[14,28]. The absence of a well-defined seed range is in agreement with structures of Type I-E Cascade complexes[26,48,50] where specialized motives in the PAM-proximal region of the R-loop were not observed. Generally, seed regions for RNA-guided nucleic acid recognition can be structurally determined, e.g., by a specific pre-ordering of the RNA-guide in the PAM-proximal region as observed for Cas9[51] and Cas12a[52]. In contrast, the guide RNA of Cascade appears to be ordered throughout its length[53] thus enabling an 'extended seed'.

The supercoil- and position-dependence of the mismatch impact directly affects the specificity of the target recognition process. CRISPR-Cas effector complexes have predominantly evolved for activity on negatively supercoiled DNA as typically found in prokaryotic cells[54]. These conditions would provide less stringent specificities but accelerated target recognition kinetics. In genome engineering applications of eukaryotic cells, however, only moderate supercoiling levels are present, which would make the target recognition more specific but also slower. Noticeably, the supercoiling in eukaryotic cells is highly locus specific[36,37] which should be considered when developing improved locus-specific off-target predictors.

From the R-loop dynamics on targets with single mismatches we could directly obtain values for individual mismatch penalties. The position independence of the penalties suggested that the random-walk model accounted correctly for the position-dependent bias observed previously[14]. The obtained free energy penalties for the three different mismatches were about 4 $k_B T$ lower than mismatch penalties within DNA duplexes in absence of proteins[55] but had similar relative differences (Supplementary Table 1). Furthermore, their average magnitude, as well as relative order, were comparable to apparent penalties determined in high-throughput in vitro binding experiments of a thermophilic Cascade complex (Supplementary Table 1)[14]. Given the position-independence of the mismatch penalties in our experiments, we expect that they are dominated by nucleic-acid thermo-dynamics but somewhat lowered due to the hybrid nature and the enforced distorted A-form of the crRNA-target strand duplex.

Most importantly, we could directly apply the single-mismatch penalties to quantitatively predict the R-loop formation dynamics on targets with two mismatches. Particularly, the random-walk model could correctly describe the proximity-dependence, i.e., an increased inhibition of R-loop formation with decreasing mismatch distance (Fig. 6c). Given its applicability to double mismatches during R-loop

formation by Cascade as well as in protein-free strand displacement reactions[35], we expect that this type of models can be easily extended to larger numbers and any types of mismatches. For such mechanism-based off-target predictors the presented work provides a thorough validation.

Generally, there are two major challenges for the establishment of mechanism-based off-target predictors: (i) the parametrization of all possible mismatch penalties corresponding to at least 48 parameters due to 12 different mismatch types and 4 different nearest neighbor base pairs onto which a mismatch can stack[55] and (ii) the determination of more detailed free-energy landscapes particularly for effector complexes with structurally determined seed motifs that will introduce localized supercoil-independent bias. In a recent analysis of high-throughput data from in vitro target libraries[13,14,32] a random walk model was employed to infer a local free energy landscape for R-loop formation by Cas9 revealing significant local bias. Despite the usage of only a single value for all mismatch penalties, an improved off-target classifier was obtained. This strongly supports the potential of mechanism-based off-target predictors. For their further improvement, we think that a combination of single-molecule measurements as presented here and of high-throughput data will be crucial. Biophysical measurements with improved temporal resolution[56] should allow to directly infer the free energy landscapes of R-loop formation for different effector complexes. In turn, high-throughput data could subsequently be used to independently parametrize mismatch penalties using a known free energy landscape. Together, this should enable better mechanism-based prediction and thus a more rational selection of target sequences for precise programming of DNA editing tools with low off-target probabilities.

In addition to the prediction of target recognition of existing effectors, the established modeling approach can also be used to devise theoretical 'optimum free energy landscapes' of R-loop formation that support maximized targeting specificity and efficiency at the same time. This can help to evaluate novel effector complexes and rationally guide further engineering approaches of high-fidelity variants[15,16,57].

## Methods

### One-dimensional random-walk model for R-loop formation

The R-loop dynamics was modeled as a random walk on a one-dimensional 1 bp lattice within a simplified free energy landscape based on the energy parameters $\Delta G_{ini}$, $\Delta G_{bias}$, $\Delta G_{lock}$, and $\Delta G_{MM}$ as described in the main text (see also Fig. 1b). The rate model of the random walk was parameterized based on the principle of detailed balance which relates the ratio of the forward (indicated by '+') and the backward (indicated by '-') rate constants between subsequent positions $n$ and $n+1$ to the free-energy difference $G_{n+1} - G_n$ between these positions:

$$\frac{k_n^+}{k_{n+1}^-} = e^{-(G_{n+1}-G_n)/k_B T} \tag{1}$$

For the unbiased free energy landscape with $\Delta G_{bias} = 0$ (Fig. 1b, top), we assume an equal rate $k_{step}$ for all transitions between R-loop states (i.e., equal kinetic barriers) except for transitions from the $U$ (and $F$) states. This excludes any sequence dependence of the stepping rates. PAM binding is comprised in the formation step of the first R-loop base pair without a distinct kinetic barrier for dissociation from the PAM. Based on these considerations the rate for R-loop initiation is given as

$$k_0^+ = k_{step} e^{-\Delta G_{ini}/k_B T} \tag{2}$$

The dependence of R-loop initiation on the Cascade concentration $c$ was included by considering the contribution of the chemical potential of the Cascade complexes to $\Delta G_{ini}$

$$\Delta G_{ini}(c) = \Delta G_{ini}(c_0) - k_B T \ln(c/c_0) \tag{3}$$

where $\Delta G_{ini}(c_0)$ is the standard initiation penalty at a reference concentration $c_0$.

For a mismatch between crRNA and DNA target strand at position $m$, we assume that the rate limiting step for mismatch establishment is the disruption of the DNA base pair, such that it occurs at the normal R-loop extension rate $k_{m-1}^+ = k_{step}$. Detailed balance provides in this case an increased rate for R-loop retraction that eliminates the mismatch

$$k_m^- = k_{step} e^{\Delta G_{MM}/k_B T} \tag{4}$$

in agreement with the rate-limiting step being facilitated by destabilized base-pairing in the heteroduplex as also indicated by the lowered kinetic barrier at the mismatch position in Fig. 1b.

The applied negative supercoiling causes a constant bias of the free energy landscape per bp in the regime where the DNA length decreases linearly with the applied turns. The torque $\tau$, which is the quantitative parameter of how the applied supercoiling stresses the DNA helix, is set by the applied force in the magnetic tweezers experiments. It was estimated as described before[58]. The bias $\Delta G_{bias}$ per bp from the torque equals the work done against the torque

$$\Delta G_{bias} = \tau \, \Delta \varphi_{bp} \tag{5}$$

where $\Delta \varphi_{bp} = 0.515$ rad ($l \,\hat{=}\, 29.5°$) is the angle by which the DNA becomes untwisted per R-loop expansion by 1 bp[25]. Assuming that the transition barrier is centered between two subsequent R-loop positions, R-loop expansion and retraction would both be affected by half of the bias providing the following corrections of all forward and backward rates for the acting torque

$$k_n^+(\tau) = k_n^+(\tau=0) e^{-\tau \Delta \varphi_{bp}/2 k_B T} \tag{6}$$

$$k_n^-(\tau) = k_n^-(\tau=0) e^{+\tau \Delta \varphi_{bp}/2 k_B T} \tag{7}$$

with $n$ being any valid position of the free energy landscape. These definitions provide a full parameterization of the rate model that describes the random walk. Mean transition times between any starting state and any end state were calculated by solving the first passage problem for this model (see Supplementary Note 1). To this end, transmissive boundaries were placed at the positions of end states and a single particle was added to the system. Upon arrival of the particle at a transmissive boundary it was instantaneously set to the start state. The mean transition time was then calculated from the steady-state flux of the single particle inside the rate landscape (see Supplementary Note 1). For intermediate R-loop states that are dynamic and extend over several base pairs, the position with the lowest free energy was taken as the state position. For DNA targets with a continuous stretch of mismatches at the PAM-distal end, the free energy landscape was cut off at the first mismatch position corresponding to an infinite free energy at this position.

For kinetic random walk simulations of R-loop length fluctuations (see Supplementary Fig. 5), we constructed the free energy landscape for a given target and calculated for each step of the lattice the forward and backward stepping rate constants $k_n^+$ and $k_n^-$. Per simulation time step $\Delta t$, a single bp forward or backward step was taken with probability $k_n^+ \Delta t$ or $k_n^- \Delta t$, respectively. $\Delta t$ was chosen sufficiently small, such that the stepping probabilities were much smaller than one.

## DNA and proteins

The 2.1 kbp DNA constructs with additional biotin- and digoxigenin modified attachment handles at either end used in the magnetic tweezers experiments were prepared as previously described[25,59]. For each DNA target presented in this study a 73 bp blunt ended oligonucleotide duplex carrying the 35 bp long target sequence was cloned into the SmaI site of a pUC19 plasmid. From the plasmids, 2.1 kbp DNA fragments containing the targets as well as SpeI and NotI restriction enzyme sites at either end were produced by PCR (primers 1 and 2 in Supplementary Table 3). Biotinylated and digoxigenin-modified ~1.2 kbp DNA fragments were produced by PCR from pBluescript II SK+ with its multiple cloning site located approximately in the center of the fragments (primers 3 and 4 in Supplementary Table 3). The biotinylated and digoxigenin-modified fragments were digested with SpeI and NotI, respectively to yield modified ~600 bp attachment handles. Following digestion of the 2.1 kbp target fragment with both restriction enzymes, it was ligated with the handles to form the final 3.3 kbp DNA construct used in magnetic tweezer experiments. For the production of complexes with different crRNAs all spacer variants were introduced into the produced vector pACYCminCR-Eco31I/SapI through the SapI and Eco31I sites using synthetic oligonucleotides with corresponding single stranded overhangs (Supplementary Table 4)[17]. Cascade complexes with different crRNAs were expressed in *E. coli* BL21 (DE3) cells and purified as described[60] using the pACYC-minCR derivatives (CmR) instead of the pACYC plasmid with homogeneous CRISPR region pCRh.

## Magnetic tweezers experiments

R-loop formation experiments were performed in 20 mM Tris-HCl (pH 8.0), 150 mM NaCl and 0.1 mg ml⁻¹ BSA at 170 pM (for experiments described in Figs. 2 and 3) or 0.5 nM (for experiments described in Figs. 4–6) Cascade using a custom-built magnetic tweezers setup[61] at room temperature (25 °C). DNA molecules were bound to 0.5 μm streptavidin-coated magnetic beads (MasterBeads; Ademtech) and added into the antidigoxigenin-coated flow cell to form tethers at the bottom surface[59,62]. Single supercoilable molecules were selected[43]. The DNA length was determined at 120 Hz by videomicroscopy and real-time GPU-accelerated three-dimensional particle tracking[63] from the position of the magnetic bead with respect to a surface-bound non-magnetic reference bead (Dynospheres; Invitrogen). Forces were calibrated from the lateral fluctuations of the DNA-tethered magnetic beads[64]. Torque values were calculated based on previous theoretical work[58,65]. For experiments in which dynamic sampling of R-loop intermediates in absence of locking was investigated (Figs. 2, 3, and 5), Cascade was added to the flow cell and the DNA molecule was negatively supercoiled (see Supplementary Fig. 3a). The number of negative turns depended on the applied force and on the change of supercoiling following the formation of the R-loop. Generally we aimed that DNA length transitions occur around half the relaxed molecule length at the given force and thus in the linear regime of the supercoiling curve[28,43]. This way the torque on the DNA stayed approximately constant. For experiments where R-loops became locked (Figs. 4 and 6), R-loop formation was induced as described before. To remove Cascade complexes with locked R-loops, the DNA molecule was positively supercoiled to (+10–14 turns, depending on the force and on the length of the individual DNA molecule) and the force was increased to ~2–3 pN to provide a high positive torque that would 'wring out' the R-loop. R-loop dissociation was seen as a sudden DNA length increase (see Supplementary Fig. 3b–e for the full procedure). R-loop formation–dissociation cycles were constantly repeated to obtain ≥25 individual events per applied condition.

## Fluorescence bulk solution experiments

All oligonucleotides for the zero torque measurements are shown in Supplementary Table 5 and were purchased HPLC-purified from Sigma-Aldrich. Shipping concentrations of 100 μM were evaluated by measuring the absorbance at 260 nm using a P-330 NanoPhotometer (Implen). Complementary strands were then annealed at a concentration of 1 μM in buffer containing 10 mM Tris-HCl (pH of 8.0), 50 mM NaCl, and 1 mM EDTA and slow cooling from 95 to the storage temperature of 4 °C at 1 K/min.

All measurements were performed in a temperature controlled Cary Eclipse at 25 °C in 1500 μL cuvettes. Before each measurement, cuvettes were rinsed 5 times with ethanol, 5 times with mili-Q, incubated overnight in 2% Hellmanex 3 solution, and again rinsed 5 times with ethanol and 5 times with mili-Q.

In the beginning, a 1350 μL solution containing the double-stranded DNA was measured for 600 s to obtain the ground level (9/10 of mean signal amplitude). Afterwards the reaction was started by quickly adding 150 μL of solution containing Cascade. Reaction conditions were 10 nM of dsDNA and 2 nM Cascade in a buffer containing 20 mM Tris−HCl (pH of 8.0), 150 mM NaCl and 0.1 μg/μL BSA.

The negative control (no protein added) was then subtracted from the measured trajectories. The fluorescence signal was then fitted to a sum of three exponentials of the form

$$F(t) = A_1\left(1 - e^{-k_1 t}\right) + A_2\left(1 - e^{-k_2 t}\right) + A_3\left(1 - e^{-k_3 t}\right) \qquad (8)$$

As quantity of interest the mean time to overcome all three steps was taken:

$$\langle t \rangle = \frac{1}{k_1} + \frac{1}{k_2} + \frac{1}{k_3} \qquad (9)$$

## Data analysis

The time resolution of the magnetic tweezers measurements can be approximated using $\tau = SNR^2 k_B T \gamma /(\kappa \Delta z)^2$, where $SNR$ is the signal-to-noise ratio and $\Delta z$ the characteristic DNA-length changes that need to be resolved for which we assumed $SNR = 3$ and $\Delta z = 50$ nm. The spring constant of the supercoiled DNA $\kappa$ and the drag coefficient $\gamma$ of the magnetic bead for axial displacements were determined from DNA length trajectories yielding $\kappa = 1.25 \cdot 10^{-3}$ pN nm⁻¹ and $\gamma = 1.5 \cdot 10^{-5}$ pN s nm⁻¹ [64]. This provided $\tau = 140$ ms, i.e., a detection bandwith of ~7 Hz. DNA length trajectories recorded at 120 Hz were therefore smoothed with a sliding average to 7.5 Hz. Transitions between the different R-loop states to generate 2-, 3-, and 4-state approximations of the R-loop trajectories were obtained by hidden Markov modeling using the vbFRET software package[66]. For the vbFRET software package, default parameters were used except for the number of expected states that was fixed to 2, 3, 4 for targets containing no (Fig. 2), one (Fig. 3) or two (Fig. 5) internal mismatches, respectively. From the discrete-state trajectories, dwell time distributions and transition rates were extracted using MATLAB (MathWorks) and LabVIEW (National Instruments). This included the generation of cumulative dwell time distributions for individual states, which were fitted to single exponential functions to obtain mean dwell times and the corresponding transition rates. For the latter, the transition probabilities to neighboring states were correspondingly considered. For experiments in which dynamic sampling of R-loop intermediates was investigated (Figs. 2, 3, and 5), trajectories of at least 3000 s were recorded for each condition including typically ~1000 transitions. For experiments where R-loops became locked (Figs. 4 and 6), ≥25 locking events were obtained for each condition to determine mean R-loop formation times. All rate and time error bars represent the standard error of the mean. Particularly, the error of mean dwell times was calculated by dividing the mean time by the square root of the number of events. Errors for all fit parameters are given as single confidence intervals.

To verify that the temporal resolution of our nanomechanical system (bead on supercoiled DNA, see above) was sufficient to resolve

the extracted R-loop transitions, we employed Brownian dynamics simulations to simulate the magnetic tweezers measurement of dynamic R-loop sampling. We first employed kinetic random-walk simulations to simulate trajectories of the R-loop dynamics (R-loop length over time) on a one-dimensional 1 bp lattice within the corresponding free energy landscape (see model description above) using the experimentally obtained parameters (Supplementary Table 1). Using the slope of the supercoiling curve at the corresponding force, we converted the R-loop length into an expected equilibrium extension $z_{DNA}(t)$ of the DNA. We furthermore modeled the diffusive fluctuations of the magnetic bead and its response to DNA extension changes by one-dimensional Brownian dynamics simulations[64]. In brief, a deviation $\triangle z$ of the fluctuating bead from the equilibrium extension, caused a back-driving drift force $F_{drift} = -\kappa \Delta z = z_{bead} - z_{DNA}(t)$ due to the stretch elasticity $\kappa$ of supercoiled DNA (comprising components of the DNA twist elasticity and the entropic elasticity of stretched DNA). Per time increment of the simulation $\Delta t$, $F_{drift}$ caused a displacement of $\Delta z_{drift} = \Delta t \, v = \Delta t \, F_{drift}/\gamma$, with $v$ being the steady state drift velocity of the bead inside the viscous medium for low Reynolds numbers. The viscous drag coefficient $\gamma$ of a spherical particle with radius $R$ inside a medium with viscosity $\eta$ was given by the Stokes formula $\gamma = 6\pi\eta R$. Within $\Delta t$, we furthermore considered random diffusive displacements $\Delta z_{diff}$ that were drawn from a Gaussian distribution with zero mean and a variance of $\langle \Delta z_{diff}^2 \rangle = 2D \Delta t$ with $D = k_B T/\gamma$ being the diffusion coefficient for the particle. By successively updating the bead position $z_{bead}$ by a total displacement of $\Delta z_{diff} + \Delta z_{drift}$ per time increment, we obtained the magnetic bead fluctuations in response to the R-loop length fluctuations (see light red trajectory in Supplementary Fig. 5a–d and Supplementary Fig. 7). The spring constant $\kappa \sim 0.01$ pN/nm describing the stretch elasticity of supercoiled DNA as well as the effective hydrodynamic bead radius $R \sim 800$ nm were obtained from power spectral density analysis of corresponding experimental trajectories of the length fluctuations of supercoiled DNA[64].

We next identified for the simulated magnetic tweezers trajectories transitions between different R-loop intermediates using vbFRET and compared them with the actual transitions of the R-loop length simulations. We could correctly identify transitions for simple R-loop intermediates (as in Fig. 2, main text) as well as for the 3-state transitions observed for sufficiently strong single mismatches (C:C and C:T, see Fig. 3, main text, and Supplementary Fig. 5a, b). For targets containing a single C:A mismatch we observed a considerably slower collapse of the full R-loop state $F^*$ (rate $k_4$) compared to the C:T and C:C mismatches (Fig. 3b and Supplementary Fig. 5e) despite the anticipated independence of the R-loop collapse on the mismatch strength. We therefore hypothesized for this weak mismatch that transitions over the mismatch between the $I$ to the $F^*$ states (rates $k_3, k_4$) were too fast to be reliably detected given the temporal resolution of our setup. To correct for the undetected transitions between $I$ and $F^*$ states we increased the R-loop collapse rate $k_4$ to the level measured for C:C and C:T mismatches $(0.8 - 1.2\,\text{s}^{-1})$, which proportionally also increases the R-loop formation rate $k_3$ and used the adjusted rates for characterizing the mismatch penalty (see Supplementary Fig. 5e). We additionally carried out simulations using the adjusted rates. Simulations of the R-loop length fluctuations provided the expected fast transitions (see Supplementary Fig. 5e, green data points). The simulated magnetic tweezers trajectories provided significantly lower transition rates that agreed with the experimental rates, supporting a correct adjustment of the rates for this weak mismatch. The correction procedure was also applied for extracting transition rates for the dynamic sampling of R-loop intermediates in case of double mismatches (positions 11, 17, Fig. 5e and Supplementary Fig. 7). While transitions between $U$ and $I$ as well as between $I^*$ and $F^*$ were correctly reproduced by the simulations, part of the

fast transitions between the $I$ and $I^*$ states were not detected (Supplementary Fig. 7). Transition rate comparisons obtained from each trajectory are represented in Fig. 5e.

### Estimation of the DNA torque in *E. coli* cells

Typical superhelical densities $\sigma$ (i.e., the number of added superhelical turns per helical turns of the relaxed DNA) found in *E. coli* cells range from $-0.06$ to $-0.029$[46,47]. For plasmid DNA added superhelical turns are partitioned between writhe and twist at a ratio of 0.8 to 0.2[67]. Thus, the superhelical density contributing to the DNA twist is

$$\sigma_{Tw} \approx 0.2\sigma \qquad (10)$$

The torque $\tau$ in a twisted semiflexible polymer of length $L$ can be calculated from

$$\tau = \frac{k_B T \, p_{tor}}{L} \Delta\varphi = \frac{k_B T \, p_{tor}}{h} 2\pi \, \sigma_{Tw} \qquad (11)$$

where $p_{tor}$ is the torsional persistence length and $\Delta\varphi$ the twist angle. The number of helical turns within $L$ is given by $N_{helix} = L/h$, where $h = 3.5$ nm is the helical pitch of B-form DNA. The twist angle is then given by $\Delta\varphi = 2\pi \, \sigma_{Tw} N_{helix}$. Inserting these relationships in the torque equation gives

$$\tau = \frac{k_B T \, p_{tor}}{h} 2\pi \, \sigma_{Tw} \qquad (12)$$

Using $p_{tor} = 100$ nm [68–70], we get for the typical superhelical densities in *E. coli* cells torques $\tau$ between $-8.9$ and $-4.3$ pN nm.

### Reporting summary

Further information on research design is available in the Nature Portfolio Reporting Summary linked to this article.

## Data availability

The datasets generated during and/or analyzed during the current study are available from the corresponding author(s) upon request. Source data are provided with this paper. Minimal raw dataset required to reproduce data published in this paper is available at Zenodo database. Source data are provided with this paper.

## Code availability

The custom-made LabVIEW code for the analysis of magnetic tweezers data is available at Zenodo: https://doi.org/10.5281/zenodo.7328018.

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

## Acknowledgements

We would like to thank Xenia Auerbach for the assistance in the development of the code, used in this study. We would also like to thank Andrey Krivoy, Dominik J. Kauert, Julene Madariaga-Marcos and Kristina Kasaciunaite for valuable discussions during the preparation of this manuscript.

## Author contributions

M.R. performed single molecule studies and analyzed the data. I.S. and T.S. purified proteins used in this study. P.I. performed fluorescence measurements presented in this study and analyzed the data. F.E.K. developed single molecule data analysis software. V.S. and R.S. designed the study. All authors interpreted the results and provided comments to the manuscript. All authors contributed to the preparation of the paper.

## Funding

This work was supported by a consolidator grant of the European Research Council (GA 724863) and by the Deutsche Forschungsgemeinschaft (DFG, grant SE 1646/9-1 within priority program 2141) to R.S. Open Access funding enabled and organized by Projekt DEAL.

## Competing interests

The authors declare no competing interests.
