## [Peer Review File · Nature Communications]

A quantitative model for the dynamics of target recognition and off-target rejection by the CRISPR-Cas Cascade complexREVIEWER COMMENTS

Reviewer #1 (Remarks to the Author):

Rutkauskas et al combined their wet experiments (e.g., PNAS 2014, Cell Rep 2015) with model study (JACS 2020) to elucidate the target recognition mechanism of CRISPR-Cas type I Cascade effector complex. They performed single-molecule DNA twist measurements using magnetic tweezers, and directly measured the DNA untwisting during R-loop formation and disruption, resolving multiple R-loop intermediates on DNA targets with mismatches. The authors built unified biophysical model based on a random walk on a simplified one-dimensional energy landscape, and elegantly explained many of the Cascade behaviors. Overall, the study is well designed with solid data. This reviewer has only several minor points.

Minor points:

1. One of the important conclusions that the authors suggested is that the seed length is controlled by DNA supercoiling rather than the Cascade structure. However, the evidence is based solely on kinetics. Does the author have some idea to proof this by biochemical and/or structural biology (or other) study?

2. Related to #1, RNA-guided systems (e.g., Cas9 and Ago) are to facilitate the directional guide RNA/target DNA annealing from the seed-sequence region, and to have a pre-organized A-form RNA guide by the seed sequence region to compensate for the energy loss of DNA unwinding. In the discussion (p18 L460), the authors claim, "The absence of a well-defined seed is in agreement with structures of Type I-E Cascade complexes (ref 57, 84, 86), where specialized seed motives were not observed." However, all the cited paper (ref 57, 84, 86) show A-form structure after hybridizing of the guide RNA/target DNA, suggesting that the Cascade structure supports the A-form formation anyway. [For example, in ref#57 (Xiao Y et al., Cell 2017, p58 left half), it is mentioned that "These points are nicely illustrated in our Cascade/seed-bubble structure by the additional finding that a larger-than-expected seed sequence is examined by Cascade, through the formation of two 5-bp pseudo A-form crRNA/DNA heteroduplex."]

Could this A-form preference explain some discrepancy of the obtained data from the fitting curve, e.g., slightly different torque dependency of matching region length (Fig. 2d and 2e)?

3. Bars, representing the theoretically predicted occupancies using best-fit parameters, are missing in some histogram data (e.g., Supplementary Fig. 4a, 8a and 8b).

4. Fig. 2c and 2e suggest that k_2 (collapse rate constant of R-loop) is not well fitted at high torque condition. What kind of other factors, if any, might affect it?

5. Furthermore, in Supplementary Fig. 4d, k_2 is not well fitted overall. How does the author intend to explain this point?

6. Considering the discrepancy in Supplementary Fig. 4d, it would be better to evaluate the mismatch penalties (ΔG_{MM}) for the different mismatches other than position 17 (similar graph of Fig. 3h)

7. This reviewer appreciates very much that the model nicely fit with the experimental data. However, one of the merits of the model would be the prediction of the character. Would it be possible for the author to provide new insight using their model (Other than off-target prediction. As in the off-target prediction, the author can optimize the parameters, i.e., off-target prediction is data-driven intrinsically, not model-driven) or comment regarding this point (especially new character of Cascade complex that is difficult to find by experimentally).

8. One of the unique characters of Cascade is crRNA/target DNA pairing is disrupted by Cas7 at every

6th position. Thus, the crRNA/DNA duplexes are five 5-bp segments of heteroduplexes, which are much different from single crRNA/DNA duplex in Cas9. Would it be possible to mention something from the model regarding the different behavior of R-loop formation/collapse, if any, compare to other RNA/DNA complex? [For example, hypothetically, if there are no disrupted base-pairing in the crRNA-DNA hybrid (at 6, 12, 18, 24 and 30), what kind of character can be predicted from the author's model?]

9. For Fig.4d, several important data points for fitting are missing. (Around 6 pN nm for M7, 3.5 pN nm for M11). It is better to take these data points. From the other data set and fitting curve, it looks possible to measure.

10. The vbFRET, a hidden Markov model-based analysis package on MATLAB, was used for idealizing the traces (two-state approximation). Did the author use default parameters for fitting? Please describe the parameters (or make comments) in the Methods section (e.g., p22, L614) for clarity.

11. For a few of the distributions (e.g., Fig.3b and 3c), it seems that state information is blurred by averaging operation (original 120 Hz trace was sliding averaged to 7.5 Hz). What is the rationale that the author used 7.5 Hz, and what happen when the author changes the averaging time window to faster condition.

Reviewer #2 (Remarks to the Author):

This manuscript by Rutkauskas et al. delves deep into the biophysical mechanisms of the Cascade-induced R-loop formation during DNA interference by the Cascade. The authors employed magnetic tweezers assay whose power lies in

- a) Single-molecule detection
- b) High-accuracy measurement of the extent and lifetimes of R-loop formation, thus uncovering multiple intermediate states in R-loop formation.
- c) The above two give the power to see the effect of rationally introduced roadblocks or mismatches in the process of mismatches. The study of this effect helps us understand how Cascade does DNA interferences and how it deals with mismatches.

The key takeaways for me:

- It refines the mechanism of DNA interference by Cascade. The manuscript shows that the mechanism is much more biophysically controlled by the amount of intrinsic supercooling of the DNA and is less guided by structural and enzymatic factors.
- It pushes the envelope on how extreme you can go with biophysical measurements to understand how physical parameters guide biomolecular functions. E.g., it shows one of the rare examples of how even a subtle level of DNA supercoiling can cause substantial changes in the enzymatic processes.
- It provides the rationale for fine-tuning the control of Cascade activity, which in turn would be helpful in improving the usage of Cascade in non-native systems like eukaryotic systems.
- It helps understand the delineation of CRISPR systems into Cas9, Cas13, Cascade, and others.
- It helps measure extreme fast events like kstep through extrapolation of carefully performed experiments & their modeling.

The experiments are expertly performed and analyzed. The conclusions are justified. The manuscript is clearly written, and I am generally quite positive about accepting this manuscript into Nature Communications. I would also like to commend & congratulate the authors for these 'deep' biophysical investigations and for putting together a cohesive story. But I feel that the manuscript is a bit too technical for an average reader and focuses on elements that I think are of secondary importance at the cost of features that I think would be of primary importance. In this regard, I have the following specific suggestions.

o Authors mention that their data and its global fit to the random-walk model supports the random-walk model in multiple instances. However, I think whether Cascade-induced R-loop follows random-walk processes or not is of secondary importance. The model is just a tool to understand the underlying biology. And thus, I feel the excessive mention of the model should be reduced. The over-emphasis on the model also makes the manuscript too technical for an average reader who is unlikely to be a hardcore biophysicist.

o The figures are pretty busy. The authors choose to show good but not very pertinent data for the key takeaways. In this regard, here are my specific suggestions:

♣ In Fig. 2b, 3 example trajectories per case can be reduced to two to reduce 'busyness,' i.e., keep the top and the bottom one and remove the middle one. The top and bottom are enough to get the point across. This suggestion can be applied to similar cases elsewhere in subsequent figures.

♣ In Fig. 3d, the use of k_4 to estimate k_{step} is a bit redundant, given that you already explained that k_{step} could be estimated from k_2 .

♣ IMO, Fig. 3f, g are not critical for the main message of entire Fig. 3, which is that Gmm remains the same irrespective of mismatch position. So, 3h is supremely important. Fig. 3f, 3g can be moved to supplementary and 3h made more prominent to reduce the busyness of Fig.3

♣ In Fig 4. b, c, the trajectories are quite busy. Their main purpose is to show that the dwell time before R-loop formation is different. In this regard, only a few examples without the gray region showing the changes in DNA supercoiling are sufficient.

♣ IMO, Fig. 4. f, g are better kept as supplementary. Decreasing the busyness of 4b, c, and f,g will increase the clarity of 4h.

♣ In Fig.5, since the central figure 5f is built comparing 11-17 and 14-17, I think it may be better to remove 13-17 to the SI figure. What do you think? Plus, please show the area in 5b-c that you have zoomed in to create 5d.

Of course, the removals in main text figures I have suggested need not be completely removed from the manuscript; they can be moved to the supplementary section.

Again, the discussion is a bit too heavy on the random-walk model and its physics and a little less on the biology of CRISPR systems. E.g., In the context of their data & model, authors could explain the repetitive kinks (disrupted base-pairing in crRNA-DNA hybrid) in the R-loop of Cascade complexes present at a uniform spacing in positions 6, 12,18,24,30. How do these kinks modulate the energy landscape of the R-loop formation/propagation?

Overall, as I have stated, I thoroughly enjoyed the manuscript, and my suggestions/review is to better the presentation and broad messaging of the manuscript.

Reviewer #3 (Remarks to the Author):

In this work Rutkauskas and colleagues developed a model to describe R-loop formation by the CRISPR-Cas type I complex when hybridized to target strands with varying degree of complementarity. The mechanistic model describes target recognition and off-target rejection as a random walk and is validated with single-molecule magnetic tweezer experiments that mimic DNA supercoiling in vivo. There are already models for off-target prediction available, but the novelty of

this model is that it elucidated the energy landscape as a function of the applied supercoiling, which consequently revealed that the length of the seed is dependent on the degree of supercoiling. They could also determine the absolute free energy penalties. The skeleton of the model was adapted from earlier work by other groups, but the new finding is solid and interesting enough for publication in Nature Communications.

However, this reviewer finds it strange that they claim credits for what has been already published by other groups: "our modelling (i) provides direct evidence that R-loop expansion follows a random-walk process (ii) shows that the single-base pair stepping of R-loop expansion occurs at a sub-millisecond time scale, [...] (iv) explains the non-trivial dependence of R-loop formation on the proximity between multiple mismatches[...]" This reviewer believes that they have to refer to other papers more explicitly. This will also help readers focus on the authors' own very interesting new findings and appreciate them.

Major comments

1) Figures are clear, but every part of the manuscript is too long, which is caused by repetition, dispersed attention to many different results instead of focussing on the key results and too much irrelevant background information.

2) The authors have analysed mismatches involving cytosines and done so at a handful of specific locations. Would the authors comment on how representative these experiments are for the general dynamics of off-target binding?

3) In general the manuscript is wordy. The introduction contains information that is partially repeated and that could be omitted, such as "Class 2 CRISPR-Cas systems employ single protein effectors as exemplified by Cas9 and Cas12, complexes that were repurposed as genome editing tools in different model organisms from bacteria to human cells. Effector complexes of the Class 1 systems are arranged of multiple subunits as exemplified by the Cascade complex and recently emerged as a promising tool for genome modification." Also the results section is rather lengthy and filled with observations from earlier research which is distracting. Please have a look at the sentences referring to earlier work / existing knowledge and re-evaluate if these can be moved to the introduction, shortened or omitted.

4) In the introduction it is mentioned that the model is derived from an already existing model; however, in the results it does not become clear how the models relate to one another. Would the authors explicitly include what is adapted and what is newly done here? Also cite Behrouz Eslami-Mossallam et al, "A kinetic model improves off-target predictions and reveals the physical basis of SpCas9 fidelity, BioRxiv, 2020.

5) To summarize their work and distill the insights derived from the model, the authors could consider a final (sub)figure depicting the model with its various states, the different rates of transferring between them and how these (do not) depend on mutations / supercoiling.

6) The manuscript contains as many as 6 figures. As both figure 5 & 6 are about the effect of double mismatches, perhaps they can be merged into a single figure, with some subpanels, such as 6B, moving to the Supplementary Information.

Minor comments

P2L46: Computational tools are mentioned as useful tools to pick target sites and corresponding cRNA that will have minimal off-target binding sites. According to the authors "a considerable fraction of

them remains undiscovered by such algorithms”, but no explanation as to why this is the case is provided. By including an explanation, it becomes more convincing that this work is needed. Also as the authors do not demonstrate that their own model is a better predictor, we suggest to nuance the criticism of the current predictor models (or provide a direct comparison).

P7L188 “Kstep=1900 +- 100 s⁻¹” Please report the confidence interval for this fit, and others throughout the manuscript.

P8L195 “intermediate R-loop intermediates” Double phrasing.

P9L232-234: “As an independent test of our model, we used the best fit parameters describing the R-loop dynamics to calculate the expected occupancies of the U,I and F* states and observed good agreement with the experimental data”. If these parameters were fit on experimental data, would it be fair to call this an independent test of the model?

P9-10L241-243 “Recording and fitting the torque dependence of the different transition rates for these complexes revealed that ΔG_{MM} was within error invariant for mismatch positions from 11 to 17 bp that could be experimentally accessed”. Does this mean that nucleic acid thermodynamics dominate, as stated two sentences before? If so, please say so explicitly. Also, to assess the accuracy of the calculated free energy values, would the authors report numbers for ΔG_{MM} previously found by others in comparable assays?

P11L267 & P15L380 “Error bars in all subplots correspond to SEM. n. s. – no significance.” Please include the threshold p-value for a difference to be considered significant.

P14L357-358 “We obtained agreement of the extracted rates with the model predictions and the results of Brownian dynamics simulations”. However looking at figure 5E, the difference between the theoretical prediction with the experimental observation/simulations is markedly larger than between the experiments and simulations and it could be argued that they are not in agreement. At a minimum the authors should provide an explanation for the difference in rates between theory and experiments/simulations for the I → I*, I* → I and I* → F* transitions.

Reviewer #1 (Remarks to the Author):

Rutkauskas et al combined their wet experiments (e.g., PNAS 2014, Cell Rep 2015) with model study (JACS 2020) to elucidate the target recognition mechanism of CRISPR-Cas type I Cascade effector complex. They performed single-molecule DNA twist measurements using magnetic tweezers, and directly measured the DNA untwisting during R-loop formation and disruption, resolving multiple R-loop intermediates on DNA targets with mismatches. The authors built unified biophysical model based on a random walk on a simplified one-dimensional energy landscape, and elegantly explained many of the Cascade behaviors. Overall, the study is well designed with solid data. This reviewer has only several minor points.

Minor points:

1. One of the important conclusions that the authors suggested is that the seed length is controlled by DNA supercoiling rather than the Cascade structure. However, the evidence is based solely on kinetics. Does the author have some idea to proof this by biochemical and/or structural biology (or other) study?

In the CRISPR-field the existence of a seed region for a particular effector complex is typically concluded by the observation that mismatches in a certain PAM-proximal region of the target affect either R-loop formation, DNA cleavage or interference more than PAM-distal mismatches outside of this region. Partially, also a structural basis can be identified for seed regions such as a pre-organized A-form guide RNA as found for Cas9 and Cas12a. However, there is in our opinion not a clear definition what makes up a 'proper' seed. Generally, the idea behind the seed-concept is that the R-loop has to expand over a minimum length (the seed region) in order to become sufficiently stable for further expansion/growth.

Our experimental observations clearly show that in presence of supercoiling PAM-proximal mismatches affect the R-loop formation kinetics stronger than PAM-distal mismatches in agreement with previous observations that concluded a seed region for Cascade (e.g. Semenova et al. 2011 and Fineran et al. 2014). Beyond this, we show that the length of the region over which PAM-proximal mismatches affect the R-loop formation is highly supercoil-dependent. In fact, in absence of supercoiling mismatches have an equally strong impact almost until the very end of the protospacer. In our opinion this significantly supports the absence of an intrinsic seed region but rather the control of an apparent seed length by supercoiling based on the seed definition above.

Of course, one can try to find further support for our experimental observations, e.g. using different biochemical readouts as suggested by the reviewer. In fact, we previously showed on a qualitative level that Cas3-mediated DNA cleavage is in absence of supercoiling similarly affected by a PAM-proximal and a PAM-distal mismatch (Rutkauskas et al. 2015, Fig. 4) in agreement with our R-loop formation data. We already referred in this context to our previous work (Rutkauskas et al. 2015) as well as to other studies performed *in vivo* in the Discussion of the manuscript on page 17.

Given point 2 raised by the reviewer, we hypothesize that the reviewer would ideally like to see a detection of a metastable R-loop intermediate as observed in a structure by Xiao Y et al. 2017. We would like to refer the reviewer to our answer to point 2, where we propose a 6-bp periodic modulation of the energy landscape that may provide meta-stable kinetic R-loop intermediates every 6 bps. We point out however that such R-loop intermediates must be short lived since they were not detected within the time resolution of our instrument of ~100 ms.

2. Related to #1, RNA-guided systems (e.g., Cas9 and Ago) are to facilitate the directional guide RNA/target DNA annealing from the seed-sequence region, and to have a pre-organized A-form RNA guide by the seed sequence region to compensate for the energy loss of DNA unwinding. In the discussion (p18 L460), the authors claim, "The absence of a well-defined seed is in agreement with structures of Type I-E Cascade complexes (ref 57, 84, 86), where specialized seed motives

were not observed.” However, all the cited paper (ref 57, 84, 86) show A-form structure after hybridizing of the guide RNA/target DNA, suggesting that the Cascade structure supports the A-form formation anyway. [For example, in ref#57 (Xiao Y et al., Cell 2017, p58 left half), it is mentioned that “These points are nicely illustrated in our Cascade/seed-bubble structure by the additional finding that a larger-than-expected seed sequence is examined by Cascade, through the formation of two 5-bp pseudo A-form crRNA/DNA heteroduplex.”] Could this A-form preference explain some discrepancy of the obtained data from the fitting curve, e.g., slightly different torque dependency of matching region length (Fig. 2d and 2e)?

The reviewer raises important questions with respect to a seed region. We generally agree with the reviewer that interactions of the Cascade proteins with the DNA strands will facilitate R-loop formation at least in order to compensate for the energetic penalties introduced by the disrupted base pairing in the crRNA-DNA hybrid every 6 bp. However, all the 5-bp duplex segments of the hybrid adopt a highly similar distorted A-form structure (see e.g. Mulepati et al. 2014), suggesting that they may experience a similar stabilization along the entire R-loop. Such a homogenous stabilization would change the overall intrinsic bias of the energy landscape but would barely affect the torque-and length-dependence of R-loop formation and collapse.

When speaking about a structurally-determined seed, one would like to see specific structural motives that would bind the first two PAM-proximal 5-bp segments stronger than the PAM-distal ones. Initial R-loop nucleation within such a seed would then be energetically favoured over further R-loop expansion. As pointed out by the reviewer, Cas9 and Ago promote a structural seed region using a pre-organized RNA-guide. For Cascade, two of our observations argue however against such an equivalent: (i) The length dependence of R-loop collapse is quite well described by our seed-free model. In contrast, in presence of a structural seed, elongation of the R-loop beyond the seed region would only little increase the R-loop stability. (ii) In absence of supercoiling “high-impact”-mismatches extend almost to the PAM-distal end, which is in contradiction to a localized PAM-proximal region that stabilizes the R-loop.

We have been aware about the beautiful structure of a partial R-loop of Xiao et al. 2017. It was formed at low temperature where one can expect that it represents a longer-lived kinetic intermediate on the way to full R-loop formation. Xiao et al. also proposed in their study that a dwell of the PAM-distal DNA duplex on the K-vise motives of the Cas7 backbone may promote an R-loop expansion in 6-bp intermediate. Within our model this would be reflected by a 6-bp periodic modulation of our energy landscape. While the modelling of our data clearly supports a rather homogenous global bias of the energy landscape, it would not be sensitive enough to reveal a mild modulation with 6-bp periodicity if smaller than the total bias. Likely, part of the deviations between our data and the minimalistic model with only few parameters will arise from such a modulation.

To address the comments of the reviewer, we added a new paragraph to the Discussion on page 16 to discuss a likely 6-bp modulation of the free energy landscape as well as the limits of our approach to detect these. Furthermore, we discuss the available structural data more specifically and suggest how metastable partial R-loop structures could result from a modulated free energy landscape.

3. Bars, representing the theoretically predicted occupancies using best-fit parameters, are missing in some histogram data (e.g., Supplementary Fig. 4a, 8a and 8b).

Bars, representing the theoretically predicted occupancies using best-fit parameters were added to Supplementary Fig. 4a (Supplementary Fig. 4d in current version of the manuscript). We removed panels representing occupancies in Supplementary Fig. 8a and 8b because they are not meaningful in case of locked R-loop formation since the considered states are not in equilibrium with each other.

4. Fig. 2c and 2e suggest that k_2 (collapse rate constant of R-loop) is not well fitted at high torque condition. What kind of other factors, if any, might affect it?

The fit of the data in Figure 2e (occupancies in Figures 2b,c were calculated from these parameters) is a single global fit to all shown curves using a single fit parameter, which is the single-bp stepping rate k_{step} . k_{step} serves essentially as a simple multiplication factor for all curves to match the absolute rate values. In the shown semilogarithmic plots, changes to k_{step} would solely shift the curves by the same amount up or down without changing their slopes nor their relative distances. We were actually quite happy to get such a good agreement with the data over two orders of magnitude in which the slopes of the curves are intrinsic predictions of the model. As discussed in point 2, slight (periodic) modulations or deviations from a constant bias might somewhat alter the shape of the curves as well as their relative distances. Also, we infer the torque from our force calibrations which are also subject to measurement error.

As pointed out in response #2, we added a paragraph to the Discussion on page 16 where we suggest that (periodic) modulations in the energy landscape may lead to the deviations from our data. Also, we emphasize that direct determination of the energy landscape of R-loop formation would be important to further refine mechanism-based off-target models.

5. Furthermore, in Supplementary Fig. 4d, k_2 is not well fitted overall. How does the author intend to explain this point?

As detailed in response #4, our model uses a minimal parameter set including a simplified energy landscape of R-loop formation. Some deviations between model and data do therefore occur as discussed in the point before likely due to modulations of the energy landscape. The data in Supplementary Fig. 4d (Supplementary Fig. 4g in the current version of the manuscript) was obtained for a mismatch at position 14. Compared to the data in Fig. 3d, k_2 increased by almost 2 orders of magnitude. This large change is rather well described by the model within relative deviations of a factor of 2. A factor of 2 corresponds to activation energy differences of only $0.7 k_B T$, which is considerably smaller than the free energy reduction at position 14 due to the applied torque (about -5 pN nm) of $\sim 8.5 k_B T$. We would like to refer the reviewer to the newly added paragraph (see above), where we discuss the possible origin for deviations between model and data.

6. Considering the discrepancy in Supplementary Fig. 4d, it would be better to evaluate the mismatch penalties (ΔG_{MM}) for the different mismatches other than position 17 (similar graph of Fig. 3h)

We provide in our manuscript a detailed position dependence for the C:C mismatch in between the same neighbouring base pairs. Testing the position dependence for 'weaker' mismatches becomes difficult for the more PAM-proximal mismatch positions, since k_2 (see Fig 3d in the current version of the manuscript) and k_3 (weaker mismatch) strongly increase. This makes the I state rather short-lived such that obtained mismatch penalties become much more error-prone than the C:C mismatch. We think that the characterization of weaker mismatches at different positions should be done using high-throughput measurements, which are beyond the scope of the study.

7. This reviewer appreciates very much that the model nicely fit with the experimental data. However, one of the merits of the model would be the prediction of the character. Would it be possible for the author to provide new insight using their model (Other than off-target prediction. As in the off-target prediction, the author can optimize the parameters, i.e., off-target prediction is data-driven intrinsically, not model-driven) or comment regarding this point (especially new character of Cascade complex that is difficult to find by experimentally).

We disagree somewhat with the idea of the reviewer that off-target prediction within our model is only data-driven. Our model *intrinsically* includes and predicts several dependencies, such as a position-dependence of the mismatch impact, the dependence on supercoiling and the mismatch proximity which were not parametrized at all. They solely arise from the random-walk nature of R-loop formation. The aim of the manuscript was to show that R-loop formations follows the predictions of the model that was based on very few and simple assumptions/parameters.

Furthermore, mismatch penalties are physically justified and measurable parameters and therefore different to heuristic scoring parameters.

Nonetheless, the reviewer raises an important point, since model-based predictions can inspire and open up new directions rather than just describing experimental observations. Since our model can predict the target recognition specificity based on an intrinsic energy landscape of R-loop formation, it could be employed to find theoretical energy landscapes that support an optimized specificity and at the same time an efficient R-loop formation on matching sequences. This would provide a helpful basis to rationally identify and engineer optimized effector complexes and to better understand the specificity of these complexes.

We added this perspective to the Discussion.

8. One of the unique characters of Cascade is crRNA/target DNA pairing is disrupted by Cas7 at every 6th position. Thus, the crRNA/DNA duplexes are five 5-bp segments of heteroduplexes, which are much different from single crRNA/DNA duplex in Cas9. Would it be possible to mention something from the model regarding the different behavior of R-loop formation/collapse, if any, compare to other RNA/DNA complex? [For example, hypothetically, if there are no disrupted base-pairing in the crRNA-DNA hybrid (at 6, 12, 18, 24 and 30), what kind of character can be predicted from the author's model?]

As explained above, we now provide more detail on the global bias of the free energy landscape of Cascade and its likely 6-bp modulation due to the disrupted base-pairing. For other effector complexes particularly with structurally determined seeds (Cas9, Cas12a), we expect pronounced local bias/local minima in the free energy landscape. We discuss this at the end of the Discussion (pages 18/19). Beyond this we would not like to speculate, since these energy landscapes may depend strongly on local motives.

9. For Fig.4d, several important data points for fitting are missing. (Around 6 pN nm for M7, 3.5 pN nm for M11). It is better to take these data points. From the other data set and fitting curve, it looks possible to measure.

We measured additional data points for M7 mismatch at a torque of -4.4 pN nm and M11 mismatch at the torque of -3.8 pN nm. The completed data set is shown in the Fig. 4b in the current manuscript version. R-loop formation data was refitted and changed values were corrected in the manuscript. Please note that for the M7 data points at torque values of -6.7 pN nm, -7.1 pN nm and -7.5 pN nm are present in the plot although covered by data points of other mismatches. They are more visible in the SI Fig. 6d in semi-logarithmic representation.

10. The vbFRET, a hidden Markov model-based analysis package on MATLAB, was used for idealizing the traces (two-state approximation). Did the author use default parameters for fitting? Please describe the parameters (or make comments) in the Methods section (e.g., p22, L614) for clarity.

The only parameter that was modified from the default vbFRET package parameters was the number of expected states that was fixed to 2, 3 or 4 for targets containing only terminal mismatches (Fig. 2), single internal and 6 terminal mismatches (Fig. 3) and two internal mismatches and 6 terminal mismatches (Fig. 5) respectively. This information is now provided in the Methods section.

11. For a few of the distributions (e.g., Fig.3b and 3c), it seems that state information is blurred by averaging operation (original 120 Hz trace was sliding averaged to 7.5 Hz). What is the rationale that the author used 7.5 Hz, and what happen when the author changes the averaging time window to faster condition.

The length change Δz that can still be resolved with a time resolution of τ is given by:

$$\Delta z = SNR \sigma_z(\tau) \approx SNR \sqrt{\frac{k_B T \gamma}{\kappa^2 \tau}}$$

where SNR is the signal to noise ratio, $\sigma_z(\tau)$ is the RMS noise when averaging the signal with a sliding window of length τ , γ the drag coefficient of the bead and κ the stiffness of the system in axial direction. From a power-spectral-density analysis (Daldrop et al. 2015), we determined from the trajectories of the DNA length of the supercoiled DNA $\kappa = 1.25 \cdot 10^{-3}$ pN nm⁻¹ and $\gamma = 1.5 \cdot 10^{-5}$ pN s nm⁻¹. Using an $SNR = 3$ and a characteristic scale of DNA length changes between states of $\Delta z = 50$ nm, we obtain a time resolution of:

$$\tau = SNR^2 \frac{k_B T \gamma}{\kappa^2 \Delta z^2} = 140 \text{ ms}$$

which agrees with an averaging frequency of 7.5 Hz. Our estimate of the best filter frequency was empirically supported by the observation that less smoothing, i.e. higher averaging frequencies did not allow to reliably determine the states of a trajectory. At higher frequencies the trajectory is still affected by correlated noise due to the limited diffusion of the magnetic bead, which in our opinion is misinterpreted by vbFRET. As an additional test, we performed simulation of the magnetic tweezers trajectories (explained in Methods and SI Fig. 5) that confirmed that the applied averaging did not contribute to the loss of short lived events in the vbFRET analysis. We added a small part under Data analysis in the Methods section to provide the time resolution of our measurements.

Reviewer #2 (Remarks to the Author):

This manuscript by Rutkauskas et al. delves deep into the biophysical mechanisms of the Cascade-induced R-loop formation during DNA interference by the Cascade. The authors employed magnetic tweezers assay whose power lies in

a) Single-molecule detection

b) High-accuracy measurement of the extent and lifetimes of R-loop formation, thus uncovering multiple intermediate states in R-loop formation.

c) The above two give the power to see the effect of rationally introduced roadblocks or mismatches in the process of mismatches. The study of this effect helps us understand how Cascade does DNA interferences and how it deals with mismatches.

The key takeaways for me:

- It refines the mechanism of DNA interference by Cascade. The manuscript shows that the mechanism is much more biophysically controlled by the amount of intrinsic supercooling of the DNA and is less guided by structural and enzymatic factors.
- It pushes the envelope on how extreme you can go with biophysical measurements to understand how physical parameters guide biomolecular functions. E.g., it shows one of the rare examples of how even a subtle level of DNA supercoiling can cause substantial changes in the enzymatic processes.
- It provides the rationale for fine-tuning the control of Cascade activity, which in turn would be helpful in improving the usage of Cascade in non-native systems like eukaryotic systems.
- It helps understand the delineation of CRISPR systems into Cas9, Cas13, Cascade, and others.
- It helps measure extreme fast events like kstep through extrapolation of carefully performed experiments & their modeling.

The experiments are expertly performed and analyzed. The conclusions are justified. The manuscript is clearly written, and I am generally quite positive about accepting this manuscript into Nature Communications. I would also like to commend & congratulate the authors for these 'deep' biophysical investigations and for putting together a cohesive story. But I feel that the manuscript is a bit too technical for an average reader and focuses on elements that I think are of secondary

importance at the cost of features that I think would be of primary importance. In this regard, I have the following specific suggestions:

1) Authors mention that their data and its global fit to the random-walk model supports the random-walk model in multiple instances. However, I think whether Cascade-induced R-loop follows random-walk processes or not is of secondary importance. The model is just a tool to understand the underlying biology. And thus, I feel the excessive mention of the model should be reduced. The over-emphasis on the model also makes the manuscript too technical for an average reader who is unlikely to be a hardcore biophysicist.

On the one hand we fully understand the concerns of the reviewer with respect to the general audience. On the other hand, the primary aim of the manuscript is to demonstrate that a rather simple random walk can quantitatively describe the R-loop/targeting dynamics as well as to reproduce/explain previous biochemical observations (e.g. seed dependence). To find a better balance between the two aspects, we reduced some of the mentioning and explanations with respect to model and energy landscape (e.g. on p. 11, p. 14, p. 16). We also modified and shortened the first part of the Discussion and removed further down additional model explanations. We hope that these changes simplify the understanding for the general reader. Also, we hope that the added Discussion sections on a likely periodic modulation of the free energy landscape, its consequences on R-loop intermediates and the identification of an energy landscape with optimum targeting specificity will further increase the general clarity of our text.

2) The figures are pretty busy. The authors choose to show good but not very pertinent data for the key takeaways. In this regard, here are my specific suggestions:

♣ In Fig. 2b, 3 example trajectories per case can be reduced to two to reduce 'busyness,' i.e., keep the top and the bottom one and remove the middle one. The top and bottom are enough to get the point across. This suggestion can be applied to similar cases elsewhere in subsequent figures.

We followed the suggestion of the reviewer and reduced the number of trajectories in Fig. 2b and 2c. We also moved the torque-dependent trajectories from Figs 3c and 5c to the Supplementary Figs. 4a and 7c, respectively. We retained the 3 trajectories for the different mismatches in Fig. 3b and Fig. 5b to intuitively show the gradual decrease of the occupancy of the F^* state with decreasing distance between mismatches.

♣ In Fig. 3d, the use of k_4 to estimate k_{step} is a bit redundant, given that you already explained that k_{step} could be estimated from k_2 .

For a given mismatch type, we applied a global fit to all four rates and their torque dependencies at once as specified in main text and the figure captions. The shown fit parameters are predominantly determined by particular rates. While ΔG_{ini} and ΔG_{MM} are mainly determined by a single rate, k_{step} mainly determined by the two highly similar collapse processes.

♣ IMO, Fig. 3f, g are not critical for the main message of entire Fig. 3, which is that G_{mm} remains the same irrespective of mismatch position. So, 3h is supremely important. Fig. 3f, 3g can be moved to supplementary and 3h made more prominent to reduce the busyness of Fig.3

We followed the suggestion of the reviewer and moved Figs. 3f and 3g to the Supplementary Figs. 4b and 4c. Fig. 3h (now Fig. 3c) was shown more prominently. Additionally, in order to reduce the busyness of Fig. 3, we moved Fig. 3c to Supplementary Fig. 4a (see also point above).

♣ In Fig 4. b, c, the trajectories are quite busy. Their main purpose is to show that the dwell time before R-loop formation is different. In this regard, only a few examples without the gray region showing the changes in DNA supercoiling are sufficient.

We followed the general suggestions of the reviewer to reduce the busyness of the figure and completely moved Figs. 4b and 4c with their mismatch position- and torque-dependent trajectories to Supplementary Fig. 6a. Gray regions showing DNA supercoiling were removed.

♣ IMO, Fig. 4. f, g are better kept as supplementary. Decreasing the busyness of 4b, c, and f,g will increase the clarity of 4h.

We followed the suggestion of the reviewer and moved Fig. 4f and 4g to Supplementary Figs 6e and 6f, respectively.

♣ In Fig.5, since the central figure 5f is built comparing 11-17 and 14-17, I think it may be better to remove 13-17 to the SI figure. What do you think? Plus, please show the area in 5b-c that you have zoomed in to create 5d.

For consistency and also to show the gradual decrease of the F^* state occupancy with decreasing distance between mismatches, we maintained all three trajectories. Instead, in order to simplify the figure, we moved Figs. 5c and 5e to Supplementary Figs. 7c and 7d (as mentioned above), removed Fig. 5d and extended Fig. 5b by the zoomed regions that are also indicated by dashed lines.

3) Of course, the removals in main text figures I have suggested need not be completely removed from the manuscript; they can be moved to the supplementary section.

Again, the discussion is a bit too heavy on the random-walk model and its physics and a little less on the biology of CRISPR systems. E.g., In the context of their data & model, authors could explain the repetitive kinks (disrupted base-pairing in crRNA-DNA hybrid) in the R-loop of Cascade complexes present at a uniform spacing in positions 6, 12,18,24,30. How do these kinks modulate the energy landscape of the R-loop formation/propagation?

We followed the recommendation of the reviewer and moved the above mentioned figure parts to the Supplementary section. In the response to this reviewer but also to reviewer 1 we added a rather extensive part to the Discussion in which we propose that the globally biased energy landscape of Cascade should have a mild 6-bp periodic modulation due to the periodically disrupted base pairing. This modulation would support the formation of metastable R-loop intermediates every 6 bp. Also we discuss in more detail the proposed absence of a structurally determined seed motif for Cascade and how this differs from other effector complexes. We hope that this provides a better link between model interpretation and biology of CRISPR-Cas systems.

4) Overall, as I have stated, I thoroughly enjoyed the manuscript, and my suggestions/review is to better the presentation and broad messaging of the manuscript.

We thank the reviewer for his/her positive and helpful comments. As detailed above, we tried in our revision to focus more on structural and biological aspects of the Cascade system and also to reduce the complexity of the manuscript (see also comments by reviewer #3)

Reviewer #3 (Remarks to the Author):

In this work Rutkauskas and colleagues developed a model to describe R-loop formation by the CRISPR-Cas type I complex when hybridized to target strands with varying degree of complementarity. The mechanistic model describes target recognition and off-target rejection as a random walk and is validated with single-molecule magnetic tweezer experiments that mimic DNA supercoiling in vivo. There are already models for off-target prediction available, but the novelty of this model is that it elucidated the energy landscape as a function of the applied supercoiling, which consequently revealed that the length of the seed is dependent on the degree of supercoiling. They could also determine the absolute free energy penalties. The skeleton of the model was adapted from earlier work by other groups, but the new finding is solid and interesting enough for publication in Nature Communications.

However, this reviewer finds it strange that they claim credits for what has been already published by other groups: "our modelling (i) provides direct evidence that R-loop expansion follows a random-walk process (ii) shows that the single-base pair stepping of R-loop expansion occurs at a sub-millisecond time scale, [...] (iv) explains the non-trivial dependence of R-loop formation on the proximity between multiple mismatches[...]" This reviewer believes that they have to refer to other papers more explicitly. This will also help readers focus on the authors' own very interesting new findings and appreciate them.

Major comments

1) Figures are clear, but every part of the manuscript is too long, which is caused by repetition, dispersed attention to many different results instead of focussing on the key results and too much irrelevant background information.

Throughout the manuscript we tried to shorten the text and focus more on the story line (see also responses to other points below). This includes shortening and focusing the introduction at several locations, shortening the first results section, removal of repetitive statements of "support for the random walk-model", removal of some energy landscape-based explanations of the obtained results, shortening and substantial rewriting of the second paragraph of the Discussion. We furthermore reduced the complexity of most figures by moving sub-panels to the Supplementary section.

2) The authors have analysed mismatches involving cytosines and done so at a handful of specific locations. Would the authors comment on how representative these experiments are for the general dynamics of off-target binding?

Generally, our data supports the idea that mismatches add only a strongly localized penalty to the free energy landscape, such that the R-loop expansion/shortening is also only locally affected. We therefore do not expect larger conceptual differences for other mismatch types. In fact, previous less qualitative data on single mismatches (Rutkauskas et al. 2015) included additionally T:T as well as A:A mismatches for which we obtained similar R-loop formation trajectories including a pronounced intermediate state. For a precise determination of mismatch penalties, we decided to characterize the full state dynamics for each mismatch type and position as function of torque. Such experiments are somewhat tedious and inherently low throughput. We therefore focused on cytosine mismatches since for these we could resolve R-loop intermediates for all three possible mismatch combinations (i.e. sufficiently strong mismatches).

To scan a large number of different mismatches, we propose now in more detail to employ data from high-throughput binding studies and highlight in this context the recent work of Behrouz Eslami-Mossallam et al. 2022. We think that biophysical experiments as done in our study are particularly useful to test and determine free energy landscapes of R-loop formation while characterization of the many different mismatch penalties should be better performed by high-throughput studies.

3) In general the manuscript is wordy. The introduction contains information that is partially repeated and that could be omitted, such as "Class 2 CRISPR-Cas systems employ single protein effectors as exemplified by Cas9 and Cas12, complexes that were repurposed as genome editing tools in different model organisms from bacteria to human cells. Effector complexes of the Class 1 systems are arranged of multiple subunits as exemplified by the Cascade complex and recently emerged as a promising tool for genome modification." Also the results section is rather lengthy and filled with observations from earlier research which is distracting. Please have a look at the sentences referring to earlier work / existing knowledge and re-evaluate if these can be moved to the introduction, shortened or omitted.

We considerably shortened the mentioned part in the introduction, simplified the explanation of the targeting mechanism in the introduction and the results and tried to avoid too extensive referring to earlier work (see also response to point 1).

4) In the introduction it is mentioned that the model is derived from an already existing model; however, in the results it does not become clear how the models relate to one another. Would the authors explicitly include what is adapted and what is newly done here? Also cite Behrouz Eslami-Mossallam et al, "A kinetic model improves off-target predictions and reveals the physical basis of SpCas9 fidelity, BioRxiv, 2020.

We refer more clearly to previous models in the introduction and particularly at the beginning of the Results section. We made clear that differences between the models are mainly due to the employed free energy landscapes and specify in the following the free energy landscape of Cascade. We already cited the nice work of Behrouz Eslami-Mossallam et al. in our previous manuscript version and so do now at multiple locations. We now explicitly highlight this work at the end of the Discussion as a first improved mechanism-based off-target predictor.

5) To summarize their work and distill the insights derived from the model, the authors could consider a final (sub)figure depicting the model with its various states, the different rates of transferring between them and how these (do not) depend on mutations / supercoiling. Though we very much agree with the reviewer that a summarizing model would be good to include, we could not imagine a good implementation without that it becomes too complex. Since the manuscript is already quite dense and includes cartoon free energy landscapes as well as example trajectories with cartoons of the different states at multiple instances we did not include a summarizing scheme.

6) The manuscript contains as many as 6 figures. As both figure 5 & 6 are about the effect of double mismatches, perhaps they can be merged into a single figure, with some subpanels, such as 6B, moving to the Supplementary Information.

We maintained Figures 5 and 6 separate. Our rationale is that each figure represents separate experiment: Fig. 2 – unlocked R-loops of different length; Fig. 3 – single mismatches and unlocked R-loops; Fig. 4 – single mismatches and locked R-loops; Fig. 5 – two mismatches and unlocked R-loops; Fig. 6 – two mismatches and locked R-loops.

We followed nonetheless the suggestion of the reviewer to substantially simplify both figures and moved Figs 5c and 5e to the Supplementary Figs. 7c and 7d as well as removed Fig. 5d and extended Fig. 5b instead.

Minor comments

P2L46: Computational tools are mentioned as useful tools to pick target sites and corresponding cRNA that will have minimal off-target binding sites. According to the authors “a considerable fraction of them remains undiscovered by such algorithms”, but no explanation as to why this is the case is provided. By including an explanation, it becomes more convincing that this work is needed. Also as the authors do not demonstrate that their own model is a better predictor, we suggest to nuance the criticism of the current predictor models (or provide a direct comparison). We now specify that the algorithms often fail to predict weaker off-targets. Also, we explain that they cannot predict how off-targeting changes with altered conditions, such as the local genomic supercoiling or the enzyme concentration.

Furthermore, we now refer directly to the recent work of Behrouz Eslami-Mossallam et al. in the Discussion as a first example for an improved mechanism-based off-target predictor. We think that this will substantiate our statement that model-based predictors may allow significant improvements.

P7L188 “Kstep=1900 +- 100 s-1” Please report the confidence interval for this fit, and others throughout the manuscript.

For all fits single confidence intervals were reported, which is now mentioned under “Data analysis” in the methods section.

P8L195 “intermediate R-loop intermediates” Double phrasing.
This was corrected as indicated.

P9L232-234: “As an independent test of our model, we used the best fit parameters describing the R-loop dynamics to calculate the expected occupancies of the U,I and F^* states and observed good agreement with the experimental data”. If these parameters were fit on experimental data, would it be fair to call this an independent test of the model?

We agree with the reviewer. It is an additional rather than independent test. We rephrased the part on page 9 and say that “Consistently, expected occupancies of the U , I and F^* calculated from the best fit parameters were also in agreement with the measurements”.

P9-10L241-243 “Recording and fitting the torque dependence of the different transition rates for these complexes revealed that ΔG_{MM} was within error invariant for mismatch positions from 11 to 17 bp that could be experimentally accessed”. Does this mean that nucleic acid thermodynamics dominate, as stated two sentences before? If so, please say so explicitly. Also, to assess the accuracy of the calculated free energy values, would the authors report numbers for ΔG_{MM} previously found by others in comparable assays?

We state now directly in the Discussion (page 17, bottom) that the penalties are generally dominated by nucleic acid thermodynamics but affected by the hybrid nature and the enforced distorted A-form of the duplex. We discuss more specifically that the obtained penalties are globally lowered by 4kBT compared to mismatch penalties of pure DNA duplexes. Furthermore, we compare them to apparent penalties from high-throughput data (Jung et al. 2017, not corrected for position bias, see Supplementary Table 1) for the same mismatch types but averaged over different nearest neighbour base-pairs. The apparent penalties had a comparable magnitude and the same order regarding mismatch strength, though the relative differences were less pronounced than in our experiments. We refer to this comparison in the same Discussion part.

P11L267 & P15L380 “Error bars in all subplots correspond to SEM. n. s. – no significance.” Please include the threshold p-value for a difference to be considered significant.

No significance was considered for $p > 0.1$, This was added to the corresponding figure captions.

P14L357-358 “We obtained agreement of the extracted rates with the model predictions and the results of Brownian dynamics simulations”. However looking at figure 5E, the difference between the theoretical prediction with the experimental observation/simulations is markedly larger than between the experiments and simulations and it could be argued that they are not in agreement. At a minimum the authors should provide an explanation for the difference in rates between theory and experiments/simulations for the $I \rightarrow I^*$, $I^* \rightarrow I$ and $I^* \rightarrow F^*$ transitions.

Analysis of the measured data misses some of the fast transitions between the I and I^* states. The actual rates are therefore somewhat underestimated and thus lower than the theory predictions. To test/verify this, we carried out the Brownian dynamics simulations using the predicted rates from the model. The good agreement between the rates from measurements and simulations supports that the theory predictions are correct. The sentence in the main text was therefore be corrected to “we obtained agreement for the extracted rates from measurements and from Brownian dynamics simulations based on the model predictions ...”. To reduce the overall complexity of the manuscript, we moved panel 5e to Supplementary Fig. 7d but still refer to it in the main text.

REVIEWERS' COMMENTS

Reviewer #1 (Remarks to the Author):

The revised paper is further strengthened and addressed my comments. It now appears suitable for publication.

Reviewer #2 (Remarks to the Author):

The authors have addressed all my comments satisfactorily. I would now recommend that the manuscript is ready for publication in Nature communications.

Best regards,

Reviewer #3 (Remarks to the Author):

The authors have made revisions which sufficiently address the previous concerns. This referee recommends the publication of this manuscript in Nature Communications.

REVIEWERS' COMMENTS

Reviewer #1 (Remarks to the Author):

The revised paper is further strengthened and addressed my comments. It now appears suitable for publication.

We are happy to hear that and thank the reviewer for previous comments and suggestions.

Reviewer #2 (Remarks to the Author):

The authors have addressed all my comments satisfactorily. I would now recommend that the manuscript is ready for publication in Nature communications.

Best regards

We are happy to hear that and thank the reviewer for previous comments and suggestions.

Reviewer #3 (Remarks to the Author):

The authors have made revisions which sufficiently address the previous concerns. This referee recommends the publication of this manuscript in Nature Communications.

We are happy to hear that and thank the reviewer for previous comments and suggestions.